

# Multi-components ensembles of future meteorological and natural snow conditions in the Northern French Alps

Deborah Verfaillie[1], Matthieu Lafaysse[1], Michel Déqué[2], Nicolas Eckert[3], Yves Lejeune[1], and Samuel Morin[1]

[1]Météo-France - CNRS, CNRM UMR 3589, Centre d'Études de la Neige, Grenoble, France
[2]Météo-France - CNRS, CNRM UMR 3589, Toulouse, France
[3]Université Grenoble Alpes, Irstea, UR ETGR, Grenoble, France

*Correspondence to:* Samuel Morin, samuel.morin@meteo.fr

**Abstract.**

This article introduces climate variations of annual-scale indicators for seasonal snow and its meteorological drivers, at 1500 m altitude in the Chartreuse mountain range in the Northern French Alps. Past and future variations were computed based on reanalysis and observations from 1958 to 2016, and using CMIP5/EURO-CORDEX GCM/RCM pairs spanning historical (1950-2005) and RCP2.6 (4), RCP4.5 and RCP8.5 (13 each) future scenarios (2006-2100). The adjusted climate model runs were used to drive the multiphysics ensemble configuration of the detailed snowpack model Crocus. Uncertainty arising from physical modeling of snow accounts for 20 % typically, although the multiphysics is likely to have a much smaller impact on trends. Ensembles of climate projections are rather similar until the middle of the 21$^{st}$ century, and all show a continuation of the ongoing reduction in mean interannual snow conditions, and maintained interannual variability. The impact of the RCP becomes significant for the second half of the 21$^{st}$ century, with overall stable conditions with RCP2.6, and continued degradation of snow conditions for RCP4.5 and 8.5, the latter leading to more frequent ephemeral snow conditions. Variations of local meteorological and snow conditions show significant correlation with global temperature variations. Global temperature levels on the order of 1.5°C above pre-industrial levels correspond to a 25 % reduction of winter mean snow depth (reference 1986-2005). Even larger reduction is expected for global temperature levels exceeding 2°C. The method can address other sectorial indicators, in the field of hydropower, mountain tourism or natural hazards.

*Copyright statement.* TEXT

# 1 Introduction

Snow on the ground is one of the most climate-sensitive components of the mountain environment. Indeed, temperature variations drive shifts of the partitioning between rain and snow precipitation, and are strongly linked with the magnitude of ablation processes (e.g. melt, sublimation). Scientific studies carried out over the past decades have demonstrated that large scale climate change has a profound impact on past and future snow conditions in alpine regions throughout the world (Martin et al.,





1994; Beniston, 1997; Mote et al., 2005; Brown and Mote, 2009; Reid et al., 2015; Marty et al., 2017b) and in particular in the European Alps (Gobiet et al., 2014).

Besides its emotional and cultural visual role in the winter mountain landscape, snow is a critical water resource component
(Bosshard et al., 2014; Lafaysse et al., 2014; Olsson et al., 2015) including hydropower (Francois et al., 2015). Furthermore, snow conditions exert major controls over winter mountain tourism (Abegg et al., 2007; Spandre et al., 2016a). Snow on the ground and its climate variations are highly relevant for mountain ecosystem functioning (Boulangeat et al., 2014; Thuiller et al., 2014) and are strongly tied with the frequency and magnitude of mountain hazards such as snow avalanches (Martin et al., 2001; Castebrunet et al., 2014) and debris flows (Jomelli et al., 2015).

While a wealth of studies have addressed, with various levels of complexity, the unequivocal projected decrease of mean multi-annual snow amount along with corresponding temperature increase predicted by all existing climate change scenarios available for the European Alps (Rousselot et al., 2012; Steger et al., 2013; Gilaberte-Burdalo et al., 2014; Gobiet et al., 2014; Schmucki et al., 2014; Piazza et al., 2014; Lafaysse et al., 2014; Marty et al., 2017a), there remains a need for quantitative and authoritative information spanning various lead times at the scale of the $21^{st}$ century appropriate for socio-economic stake-
holders at the local, regional and national scale. This originates from the unavailability hitherto of required input information as well as a suitable methodological framework to identify and convey the information to their potential users in the most relevant and appropriate way. Indeed, many existing studies addressing future snow conditions in the European Alps rely on climate scenarios which have formed the basis of the $4^{th}$ IPCC Assessment Report (AR4). While their conclusions were not contradicted by the subsequent report (IPCC, 2013, 2014a, b, c), various methodological changes and updates warrant the ne-
cessity to generate renewed estimates of the impact of future climate change on meteorological and natural snow conditions in the Alps, consistent with AR5 material and conclusions (IPCC, 2013). Firstly, IPCC global scale socio-economic/greenhouse gas emission scenarios have seen major changes from AR4 to AR5, from the SRES approach to RCP (Moss et al., 2010). Secondly, global climate models have evolved from the Coupled Model Intercomparison Project Phase 3 (CMIP3) to CMIP5 (Taylor et al., 2012) and generated novel ensembles of global climate projections (Taylor et al., 2012). Last, regional climate
model outputs have recently been generated using CMIP5 climate projections as boundary conditions, providing ensemble model runs spanning the entire chronology of climate variations using historical model runs and RCP-driven projections. This concerns the time period from 1950 to 2100 in the case of the EURO-CORDEX project (Jacob et al., 2014; Kotlarski et al., 2014). Existing recent literature addressing the impact of climate change on wintertime snow conditions has only in a few cases used these latest generation model results (Terzago et al., 2017; Frei et al., 2017).

Using latest generation climate models as input for impact assessments because there are newer is not *per se* a sufficient motivation for updating existing climate impact studies (Knutti et al., 2010). Improved methodological approaches have also the potential to lever critical limitations of existing studies. For example, several recent studies (Rousselot et al., 2012; Castebrunet et al., 2014; Schmucki et al., 2014; Marty et al., 2017a) were based on so-called, more or less sophisticated, delta-change approaches applied to meteorological conditions, employed to drive snowpack models. Using such methods, as recognized by
Marty et al. (2017a), implies "that the variability does not change over time", in particular the seasonality of meteorological conditions, such as the frequency of precipitation events and their time distribution. Indeed, such approaches consist in applying




a pre-determined variation of temperature and/or precipitation values from an observation record, based on variations computed using climate models (either global or regional). This cannot capture combined variations in temperature, precipitation and other meteorological factors, both in terms of magnitude but also seasonal variations. Given that snow conditions for a given

season depend on the unfolding of meteorological conditions driving accumulation (precipitation events) and ablation of the snowpack, realistic predictions of the impact of climate change on mountain meteorological and snow conditions should instead be based on the chronology of the climate model outputs at the daily or sub-daily time resolution. However, this requires the use of downscaling and adjustment methods operating at these time scales (Déqué, 2007; Themeßl et al., 2011; Gobiet et al., 2015), in order to bridge the elevation gap induced by the difference between the spatial resolution of the regional or global

climate model and the topography of the target area (Piazza et al., 2014), and to mitigate inevitable biases held by the raw climate model outputs (Christensen et al., 2008; Rauscher et al., 2010; Kotlarski et al., 2014). Last, solid assessment of the impact of climate change on snow conditions requires handling carefully uncertainty and variability sources, in order to provide balanced and relevant information to the end-users (Brasseur and Gallardo, 2016). This can be achieved by selecting relevant indicators along with their time and space aggregation principles, relying on ensembles addressing the largest possible range of

uncertainty and variability sources, and employing a robust statistical analysis framework, in order to not only focus on changes in mean conditions (Marty et al., 2017a), but also higher order moments of the distribution of possible futures (Vasseur et al., 2014) and the statistical significance level of the computed trends (Castebrunet et al., 2014).

In this study, we introduce recent developments in the field of climate information related to meteorological and natural snow conditions, applied to the French mountain areas. The approach draws on the use of the ADAMONT statistical adjustment

method (Verfaillie et al., 2017) applied to multiple historical (1950-2005) and future (2006-2100) EURO-CORDEX regional climate model runs spanning all relevant RCPs (RCP2.6, RCP4.5 and RCP 8.5). The 13 GCM/RCM EURO-CORDEX pairs currently available are expected to span the overall uncertainty resulting from GCM errors, RCM errors and climate internal variability. We used one of the longest meteorological reanalyses available in the French mountain regions - the SAFRAN reanalysis (Durand et al., 2009b) - as the reference observational dataset. Continuous hourly-resolution meteorological time

series derived from RCM output by the ADAMONT statistical adjustment method are then used as input of the SURFEX/ISBA-Crocus snowpack model (Vionnet et al., 2012). Its default configuration and also, for the first time to the best of our knowledge, a recently developed multiphysical ensemble system (Lafaysse et al., 2017) are used, making it possible to quantify snowpack model errors in the context of climate change impact assessment. We define a series of indicators for meteorological and natural snow conditions at the annual scale based on daily temperature, precipitation, snow depth and snow water equivalent data. The

multi-ensemble datasets are analyzed using two specific statistical frameworks, addressing either individual annual values or multi-annual averages, which provide complementary information depending on the application. While the framework developed here can be applied as such in all areas where the SAFRAN system has been implemented (Durand et al., 2009b; Maris et al., 2009; Quintana-Seguí et al., 2017), we focus in this article on results obtained for the Chartreuse massif in the Northern French Alps at an altitude of 1500 m. This altitude level is particularly sensitive to climate change (Martin et al., 1994; Rousselot et al., 2012; Steger et al., 2013; Lafaysse et al., 2014; Gobiet et al., 2015; Schmucki et al., 2014; Marty et al., 2017a) and it corresponds roughly to the setting of the mid-altitude long-term observational site Col de Porte (1325 m altitude,





45.3°N, 5.77°S), which has long been used to monitor and showcase the impact of climate change on mountain snowpack and provides appropriate observational records making it possible to place in context the modeling results.

## 2 Materials and Methods

### 2.1 Observations

This study uses meteorological data from the SAFRAN reanalysis (1958-2016, Durand et al., 2009a, b). The SAFRAN system is a regional scale meteorological downscaling and surface analysis system (Durand et al., 1993), providing hourly data of temperature, precipitation amount and phase, specific humidity, wind speed, and shortwave and longwave radiation for different regions in the French Alps but also in the French and Spanish Pyrenees and Corsica. Unlike traditional reanalyses, SAFRAN does not operate on a grid, but on mountain regions subdivided into different polygons known as massifs (Durand et al., 1993, 1999), which correspond to regions of 500 to 2,000 km$^2$ for which meteorological conditions are assumed spatially homogeneous but varying only with altitude. SAFRAN data are thus available for each massif and for elevation bands with a resolution of 300 m. While all the developments and results introduced below can be generically applied to all the French mountain regions, we focus solely, for the sake of brevity, on the Chartreuse massif at an altitude of 1500 m, on flat terrain and without accounting for specific topographical masks.

Additionally, we use long-term observations from the Col de Porte observatory (CDP, 1325 m above sea level (a.s.l.), 45.3°N, 5.77°E) located in the Chartreuse massif in the French Alps (Morin et al., 2012). Daily snow depth and meteorological measurements (temperature and precipitation) are available from 1960 to 2016. At this site, the snow season generally extends from December to April, with occasional occurrences of snowmelt and rainfall events, and usually low wind speed. Note that the Col de Porte meteorological observations are not used in the SAFRAN reanalysis.

### 2.2 Climate projections

This study uses the currently available EURO-CORDEX dataset (Jacob et al., 2014; Kotlarski et al., 2014), consisting of 6 regional climate models (RCMs) forced by 5 different global climate models (GCMs) from the CMIP5 ensemble (Taylor et al., 2012) over Europe, for the historical, RCP 2.6, RCP 4.5 and RCP 8.5 scenarios (Moss et al., 2010). Historical runs generally cover the period 1950–2005 and RCPs cover the period 2006–2100, with some exceptions due either to the availability of the RCM or of the GCM. Table 1 provides the different GCM/RCM combinations used in this study. In total, 43 different 0.11° resolution (EUR 11, ≈ 12.5 km) time series of daily minimum and maximum temperature, total precipitation, longwave and shortwave incoming radiation, zonal and meridian near-surface wind speed and specific humidity were used. In order to analyze continuous long-term series (generally from 1950 to 2100 with a few exceptions), historical (HIST) and each RCP time series were concatenated (named RCP2.6, RCP4.5 and RCP8.5 in the following). The spread of this ensemble for a given RCP is due to three distinct factors: the different responses among the GCMs to a given RCP, the different responses among RCMs to a given GCM forcing, and the internal variability of climate at different time scales affecting the response of one specific model





**Table 1.** EURO-CORDEX GCM/RCM combinations used in this study, with the time period available for each scenario. Contributing institutes are indicated inside parentheses; CLMcom: CLM Community with contributions by BTU, DWD, ETHZ, UCD,WEGC; CNRM: Météo France; IPSL-INERIS: Institut Pierre Simon Laplace, CNRS, France – Laboratoire des Sciences du Climat et de l'Environnement, IPSL, CEA/CNRS/UVSQ – Institut National de l'Environnement Industriel et des Risques, Verneuil en Halatte, France; KNMI: Royal Netherlands Meteorological Institute, Ministry of Infrastructure and the Environment; MPI-CSC: Climate Service Center (CSC), Hamburg, Germany; SMHI: Rossby Centre, Swedish Meteorological and Hydrological Institute, Norrkoping Sweden.

| RCM (Institute) | GCM | HIST | RCP 2.6 | RCP 4.5 | RCP 8.5 |
|---|---|---|---|---|---|
| CCLM 4.8.17 (CLMcom) | CNRM-CM5 | 1950–2005 | | 2006–2100 | 2006–2100 |
| | EC-EARTH | 1950–2005 | | 2006–2100 | 2006–2100 |
| | HadGEM2-ES | 1981–2005 | | 2006–2099 | 2006–2099 |
| | MPI-ESM-LR | 1950–2005 | | 2006–2100 | 2006–2100 |
| ALADIN 53 (CNRM) | CNRM-CM5 | 1950–2005 | 2006–2100 | 2006–2100 | 2006–2100 |
| WRF 3.3.1F (IPSL-INERIS) | IPSL-CM5A-MR | 1951–2005 | | 2006–2100 | 2006–2100 |
| RACMO 2.2E (KNMI) | HadGEM2-ES | 1981–2005 | 2006–2099 | 2006–2099 | 2006–2099 |
| REMO 2009 (MPI-CSC) | MPI-ESM-LR | 1950–2005 | 2006–2100 | 2006–2100 | 2006–2100 |
| RCA 4 (SMHI) | CNRM-CM5 | 1970–2005 | | 2006–2100 | 2006–2100 |
| | EC-EARTH | 1970–2005 | 2006–2100 | 2006–2100 | 2006–2100 |
| | IPSL-CM5A-MR | 1970–2005 | | 2006–2100 | 2006–2100 |
| | HadGEM2-ES | 1981–2005 | | 2006–2099 | 2006–2099 |
| | MPI-ESM-LR | 1970–2005 | | 2006–2100 | 2006–2100 |

run. As in most impact studies based on EURO-CORDEX scenarios, we assume here that the 13 GCM/RCM pairs reasonably sample the overall uncertainty resulting from these 3 sources, even though not all GCM/RCM combinations are available.

The EURO-CORDEX raw surface fields were adjusted using the ADAMONT method, which is a quantile mapping and disaggregation method taking into account weather regimes to provide multi-variable hourly adjusted climate projections (Verfaillie et al., 2017). The method uses a meteorological observational dataset at hourly time resolution (here the SAFRAN meteorological reanalysis from 1980 to 2011), and regional climate model outputs covering the geographical domain of interest (here the EURO-CORDEX dataset). Raw RCM outputs for the grid point closest to the middle of the Chartreuse massif were used (see Verfaillie et al., 2017 for details). The altitude values of the RCM grid points used range from 612 to 1085 m, with a mean value across all RCMs of 880 m. Note that Verfaillie et al. (2017) have demonstrated that the ADAMONT method provides adequate results under this setting with several hundreds of meters difference between RCM and the target altitude, and that selecting RCM grid points with a larger geographical distance but lower altitude difference does not necessarily improve the outcome of the adjustment procedure.



## 2.3 Snowpack model

We used the Crocus (Vionnet et al., 2012) unidimensional multilayer snowpack model to predict snow conditions based on meteorological input data (both reanalysis and adjusted climate projections). Crocus computes the exchanges of energy and mass between the snow surface and the atmosphere and between the snowpack and the ground underneath. It requires sub-diurnal

(ideally hourly) meteorological forcing data and is able to simulate the evolution of the snowpack over time, by accounting for several processes occurring in the snowpack, such as thermal diffusion, phase changes, metamorphism, etc. In this study, we used the ESCROC (Ensemble System CROCus) multiphysics approach described in Lafaysse et al. (2017), which consists in using multiple combinations of different physical options of the model to build an ensemble of model configurations. We specifically use ensemble $E_2$ as defined in Lafaysse et al. (2017) which includes a subset of 35 configurations selected to be

equiprobable at CDP. The spread of this ensemble has been optimized at CDP and is able to explain about 2/3 of total error in simulations driven by meteorological measurements at CDP, which is a realistic contribution of snowpack model error to the total simulation error (Raleigh et al., 2015; Lafaysse et al., 2017). An additional configuration corresponding to the default Crocus configuration run was also used, totalling 36 model configurations.

## 2.4 Indicators and post-processing

### 2.4.1 Definition of indicators

Based on meteorological and snow-related variables at daily time resolution, we computed and analyzed different indicators defined at the annual time scale, using an indicator-oriented approach described in Strasser et al. (2014). Defining "winter" as the period from December to April inclusive (5 months long), the following snow condition indicators were computed: mean winter snow depth ($\overline{SD}$), exceedance duration over a snow depth threshold for thresholds values of 5 cm, 50 cm and 1 m

($STED_5$, $STED_{50}$, $STED_{100}$, expressed in days). In terms of meteorological indicators, given the focus of the present study on wintertime processes and snow conditions, we considered mean winter temperature ($\overline{T}$), cumulated winter total (rain and snow) precipitation ($\overline{P}$) and mean winter ratio between snow and total precipitation ($\overline{R}$). Relaxing the focus on the winter time period, we also computed the maximum annual snow water equivalent ($\widehat{SWE}$) as well as the snowpack onset and melt-out dates ($SOD$ and $SMOD$), which correspond to the earliest/latest time bounds of the longest period of time with snow depth

values exceeding 5 cm, which can be interpreted as the longest period of time with continuous snow cover. These indicators are meant to represent the most significant features of natural snow on the ground at the annual scale (Schmucki et al., 2014), although they are not immediately relevant for snow conditions in ski resorts (Spandre et al., 2016a) and should not be the sole source of information to be used in this context. Figure 1 provides an overview of the snow-related indicators introduced above.





**Figure 1.** Overview of the snow-related indicators introduced in section 2.4, using an arbitrary SWE and snow depth time series over the course of a given year. Top : SWE time series, displaying the maximum value $\widehat{SWE}$. Bottom : snow depth time series, displaying graphically the related indicators. See text for details.



### 2.4.2 Statistical post-processing of indicators

The post-processing of the annual indicators was performed in two complementary ways, described below, using all available EURO-CORDEX GCM/RCM pairs and assigning them, for a given RCP, equal weight (Knutti et al., 2010), i.e. equal probability to represent past climatic conditions (historical model runs) and possible future conditions. What follows applies for each Crocus model configuration, i.e. the handling of multiphysics snowpack model data corresponds to multiple instances of similar data processing.

1. Quantiles of annual values: In this case, for a given RCP, all annual values of the indicators spanning a given time window (15 years typically) for all the corresponding GCM/RCM pairs (13 in the case of RCP4.5 and RCP8.5, 4 in the case of RCP2.6) were pooled together and quantiles of the distribution of the values (195 in the case of RCP4.5 and RCP8.5, 60 in the case of RCP2.6) were determined using a kernel smoothing approach. We computed the 5%, 17%, 50%, 83% and 95% values (Q5, Q17, Q50, Q83, Q95), consistent with IPCC (2013). This approach has the advantage of providing statistical estimates for individual annual values of the indicator, although it mixes together the effects of interannual variability and inter-model variability. The spread resulting from these two sources of uncertainty and variability can be approached in relative terms, by dividing Q83-Q17 by Q50 to form a formal equivalent to the coefficient of variation, defined using quantile values instead of mean and standard deviation (referred to as quantile-based coefficient of variation -QCV- hereafter).

2. Moments of multi-annual averages: In this case, the multi-annual average of annual indicator values (15 time windows years typically) is computed, for a given RCP, for each GCM/RCM pair. For a given RCP, mean ($\mu$) and standard deviation ($\sigma$) values are computed for the ensemble of multi-annual averages of all GCM/RCM pairs. This approach provides information on the statistical distribution of each indicator for a given RCP on a multi-annual integration perspective. In practice, we compute $\sigma' = 1.37\,\sigma$, corresponding to the 17% and 83% quantiles in the case of a normal distribution, so that this approach becomes more comparable to the quantile approach. The coefficient of variation CV can be determined as CV$= 2 \times \sigma'/\mu$, which makes it comparable with the QCV definition above.

### 2.4.3 Variations of indicator values between reference and future time periods

For both methods, results over the historical period are contextualized with temporal median or mean of the annual indicators computed for the SAFRAN-Crocus reanalysis and for observations at CDP.

The values of the post-processed indicators were computed using sliding 15-year windows spanning the entire climate dataset available, i.e. from 1950 to 2100 in the case of EURO-CORDEX data (although some GCM/RCM pairs do not span the full historical period), from 1958 to 2016 in the case the SAFRAN-Crocus reanalysis, and from 1960 to 2016 in the case of CDP observations. In order to compute variations between conditions of the recent past and future changes, the reference period 1986-2005 (Ref) was selected, which contains all (and only) historical EURO-CORDEX model runs and was used as a baseline period of the IPCC AR5. Specific values of the post-processed indicators were computed for a series of representative future 15-year time windows $t$ centered on 2030, 2050, 2070 and 2090. For the snowpack indicators, values are provided for





the reference time period as well as for the future. Relative changes were computed in the case of meteorological indicators $\overline{T}$, $\overline{P}$ and $\overline{R}$. For each GCM/RCM pair $m$ the mean value over the period 1986-2005 ($\bar{x}_0^m$) was calculated, as well as mean values for 15-year windows around each future $t$ time periods for the RCP $r$ ($\bar{x}_t^{m,r}$). For temperature and the ratio between snow and total precipitation, $\Delta_t^{m,r}$ corresponds to the difference between $\bar{x}_t^{m,r}$ and $\bar{x}_0^m$, while for precipitation, $\rho_t^{m,r}$ corresponds to the percentage increase or decrease compared to the reference period, e.g., $(1 - \bar{x}_t^{m,r}/\bar{x}_0^m) \times 100$. Finally, $\mu \pm \sigma'$ values of all $\Delta_t^{m,r}$ or $\rho_t^{m,r}$ for a given $r$ and a given $t$ were determined. These calculations were performed for each RCP using all available

GCM/RCM pairs. For the reference period 1986-2005 and future time periods, the multi-model calculations were performed using either all the GCM/RCM pairs providing RCP2.6, RCP4.5 and RCP8.5 model runs (4), or all the GCM/RCM providing RCP4.5 and RCP8.5 model runs (13).

### 2.4.4 Variations of local indicators and global air temperature between reference and future time periods

For the reference period 1986-2005 and for three 30 year periods during the $21^{st}$ century (beginning of century (BOC), 2011-
10 2040, middle of century (MOD) 2041-2070 and end of century (EOC), 2071-2100), we computed interannual mean values corresponding to a given GCM/RCM pair for the meteorological and snow indicators introduced above, for all RCPs available for a given GCM/RCM pair (either RCP4.5 and RCP8.5 only, or all three RCP2.6, RCP4.5 and RCP8.5 scenarios). For each GCM/RCM model run under each available RCP configuration, the global temperature difference between future time periods (BOC, MOC and EOC, respectively) and the pre-industrial period (1851-1880), referred to as $\Delta T_{g,BOC-PI}$, $\Delta T_{g,MOC-PI}$
and $\Delta T_{g,EOC-PI}$, respectively, for the corresponding GCM and RCP was calculated (Taylor et al., 2012). In addition, the global temperature difference was also computed between future periods (BOC, MOC and EOC) and the reference (Ref) period 1986-2005 $\Delta T_{g,BOC-Ref}$, $\Delta T_{g,MOC-Ref}$ and $\Delta T_{g,EOC-Ref}$, respectively. Based on these datasets, we computed linear regressions curves (intercept forced to 0) between interannual means of the local meteorological and snow indicators during BOC, MOC and EOC, and the corresponding global annual temperature difference between the corresponding time
period and the Ref period. Linear regressions were also computed using all future time periods together (ALL). In addition, the future values of the local meteorological and snow indicators of all future time periods were binned according to the corresponding global temperature by steps of $0.5°C$ ($\pm 0.25°C$), and the mean and standard deviation of all values within a given bin were computed.

### 2.4.5 Comparison between results of numerical simulations and observations

On the basis of the annual values of the indicators $\overline{SD}$, $\overline{T}$ and $\overline{P}$ for the time period from 1986 to 2005, statistics of the differences between reanalysis data and Col de Porte observations were computed, in terms of mean bias, root mean square deviation (RMSD) and correlation (only $\overline{T}$, $\overline{P}$). This is not meant to represent an evaluation of the SAFRAN-Crocus reanalysis, because the SAFRAN dataset used in this study was not optimized to correspond exactly to the geographical setting of the Col de Porte observation site (appropriate altitude, specific terrain masks impacting solar radiation time distribution). However,
the geographical setting of the observations and simulations are sufficiently close to each other that the two can be analyzed concurrently and provide reasonable information pertaining to the ability of the model chain to represent meteorological con-



ditions in such a mountainous area. A better statistical match between observation and reanalysis would however be expected using meteorological data more applicable to the observation configuration, which is not the purpose of this article and was addressed in previous publications (Durand et al., 2009b; Lafaysse et al., 2013).

## 3 Results

This study introduces multi-component ensembles of past and future simulations of meteorological and snow conditions in the Chartreuse mountain range in the Northern French Alps at 1500 m altitude. As described previously, simulations encompass multiple RCPs, multiple GCM/RCM pairs from the EURO-CORDEX database adjusted using the ADAMONT method, and multiple Crocus snowpack model runs using the ESCROC ensemble system. This section describes the wealth of information generated through this process, focussing on meteorological and snow indicators described previously and addressing various components of the uncertainty and variability sources affecting the simulations.

### 3.1 Full ensemble configuration and uncertainty apportionment

Figure 2 provides an overview of all sources of uncertainty and variability accounted for in this study, in terms of snow conditions (using the $\overline{SD}$ indicator as an example) for the period from 1950 to 2100. Only RCP4.5 climate projection data are considered here, for the sake of brevity. The corresponding figure for RCP8.5 is available in the Supporting Information (Fig. S1).

Figure 2a shows continuous time series of annual values of mean winter snow depth data ($\overline{SD}$), either observed or generated by the default snowpack model configuration fed by meteorological data from a reanalysis or an adjusted RCM. It highlights the significant interannual variability, both in observed, reanalyzed and climate model dataset. For the time period 1986-2005, the mean observed $\overline{SD}$ value is 0.64 m. Using the default Crocus configuration fed by the SAFRAN reanalysis at 1500 m altitude yields bias and RMSD values of annual $\overline{SD}$ values of 0.10 m and 0.18 m, respectively, against the Col de Porte observational record, which falls within the commonly accepted range of snowpack modeling errors at observing stations when models are driven by meteorological observations (Essery et al., 2013; Lafaysse et al., 2017). The mean observed $\overline{T}$ value over the same period is 0.9 °C, with bias and RMSD values of -0.1 °C and 0.6 °C, respectively, when comparing SAFRAN with the Col de Porte observational record. The coefficient of determination between SAFRAN and the observations is equal to 0.85. For $\overline{P}$, the mean observed value is 777 kg m$^{-2}$, with a bias value of 7 kg m$^{-2}$ and a RMSD value of 149 kg m$^{-2}$. The coefficient of determination is equal to 0.74. The interannual variations among GCM/RCM are only correlated between RCMs forced by the same GCM but decorrelated between the different GCMs, as expected.

Figure 2b shows, both using meteorological reanalysis and adjusted climate model data (here one given GCM/RCM pair under RCP4.5 climate conditions), the spread of $\overline{SD}$ values which can be obtained using the ESCROC $E_2$ ensemble of snowpack model configurations (Lafaysse et al., 2017). The interannual variations are highly correlated between members because they are mainly driven by the GCM/RCM used as input. The plot shows by how much the snowpack modeling uncertainty affects the results in terms of mean annual snow depth under one specific climate scenario (one RCP, one GCM/RCM pair).



Figure 2c shows the ensemble of Crocus model configurations driven by the 13 GCM/RCM pairs in the case of RCP4.5,
each GCM/RCM pair being displayed with a given color. This figure shows the large multi-components ensemble of individual
annual data which can be generated when combining all available information, which highlights the need for appropriate data
synthesis methods.

Figure 2d aims at apportioning the uncertainty components arising from GCM/RCM inter-model variability (including model
uncertainty and internal variability of climate at different time scales) and multiphysics snowpack model uncertainty. This figure
is based on the multi-annual averages approach described in Section 2.4. For this, $13 \times 35$ 15-year running average values of $\overline{SD}$
values were generated, spanning all GCM/RCM combinations for RCP4.5 (13) and multiphysics Crocus ensemble ESCROC
(35). The standard deviation of these 455 values was computed for each 15-year window, and corresponds to the total standard
deviation of the $\overline{SD}$ historical and future climate conditions. This is shown in black solid line on Figure 2d, displaying values
on the order of 0.08 to 0.11 m with decennal variability but no temporal trend from 1950 to 2100. This can be viewed as the total
quantified uncertainty level for a given RCP affecting individual values of 15-year averages of $\overline{SD}$. The snowpack multiphysics
(referred to as ESCROC) and GCM/RCM uncertainty components were computed based on a further post-processing of the 455
$\overline{SD}$ 15-year averages for each 15-year window. The ESCROC component was quantified as the mean value of the 13 values
(one for each GCM/RCM pair) of the standard deviation of the 35 multiphysics configurations. Similarly, the GCM/RCM
component was quantified as the mean value of the 35 values (one for each multiphysics configuration) of the standard deviation
of the 13 GCM/RCM pairs. Time variations of these individual values are displayed in Figure 2d. The ESCROC component
shows values ranging from 0.05 m to 0.07 m, exhibiting rather smooth variations from 1950 to 2100 and a general decreasing
trend, along with the general decreasing trend of $\overline{SD}$ over the considered time period (see below). In contrast, the GCM/RCM
component shows significant variations, with values from 0.02 m to 0.11 m. Note that the assessment of this component for the
historical period is affected by the varying number of available GCM/RCM before 1980 and by a potentially artificial reduction
of spread over the 1980-2011 calibration period of the ADAMONT statistical adjustment method. The relative proportion of
these two components was estimated as the simple ratio of the corresponding variance values to the total variance value. The
variance is used in this comparison because the variances of both factors would be additive if they were independent (the
interaction term is neglected here). It shows that the ESCROC component plays in the future period a smaller role than the
GCM/RCM component, decreasing over time. This shows that the uncertainty arising from snowpack modeling errors plays
a significant (always more than 15% of variance), although secondary role, for future climate projections. Furthermore, we
anticipate that the impact of snowpack modeling uncertainties plays an even smaller role when focusing on relative changes of
simulated snow conditions because for one given GCM/RCM the different ESCROC members are usually ranked in a similar
order all along the simulation period. For these reasons, we focus below on modeling results solely using the default Crocus
model configuration and not the multiphysics ensemble. This is further discussed in the Discussion section.





**Figure 2.** Observed and simulated time series of $\overline{SD}$. a) Continuous time series of annual values of mean winter snow depth data ($\overline{SD}$), either observed or generated by the default snowpack model configuration fed by meteorological data from a reanalysis or an adjusted RCM. b) $\overline{SD}$ values obtained using the ensemble of Crocus model configurations ESCROC. c) Ensemble of Crocus model configurations driven by the 13 GCM/RCM pairs in the case of RCP4.5, each GCM/RCM pair being displayed with a given color. d) Estimate of absolute and relative contribution of uncertainty components arising from GCM/RCM inter-model variability and multiphysics snowpack model uncertainty (ESCROC).



**Table 2.** Quantile values (Q17 = 17%, Q50 = 50%, Q83 = 83%) over 15-year windows, for the reference period 1986-2005 (Ref) in observations (OBS, only Q50), SAFRAN-Crocus (S-C, only Q50) and historical scenario (HIST, *13 GCM/RCM pairs, **4 GCM/RCM pairs corresponding to the ones in RCP2.6), and around the time slots 2030, 2050, 2070 and 2090 for each future scenario (RCP2.6: 4 pairs, RCP4.5 and RCP8.5: 13 pairs), for $\overline{SD}$, $\widehat{SWE}$ and $SOD$ - $SMOD$ (mm/dd - mm/dd; for RCPs, number of days earlier or later compared to HIST* $SOD$ and $SMOD$).

| Time slot | | $\overline{SD}$ (m) | | | $\widehat{SWE}$ (kg m$^{-2}$) | | | $SOD$ - $SMOD$ | | |
|---|---|---|---|---|---|---|---|---|---|---|
| | | Q17 | Q50 | Q83 | Q17 | Q50 | Q83 | Q17 | Q50 | Q83 |
| Ref | OBS | | 0.66 | | | | | | 12/04 - 04/24 | |
| | S-C | | 0.69 | | | 389 | | | 12/03 - 05/04 | |
| | HIST* | 0.30 | 0.63 | 1.02 | 205 | 384 | 588 | 11/16 - 05/15 | 12/09 - 04/28 | 01/06 - 04/04 |
| | HIST** | 0.27 | 0.65 | 1.04 | 193 | 395 | 602 | 11/15 - 05/18 | 12/08 - 04/29 | 01/04 - 04/01 |
| 2030 | 2.6 | 0.19 | 0.43 | 0.82 | 159 | 305 | 501 | +7  -9 | +15  -9 | +17  -3 |
| | 4.5 | 0.20 | 0.46 | 0.86 | 167 | 317 | 515 | +6  -6 | +12  -10 | +14  -14 |
| | 8.5 | 0.18 | 0.45 | 0.82 | 141 | 307 | 495 | +6  -7 | +10  -13 | +9  -31 |
| 2050 | 2.6 | 0.16 | 0.46 | 0.83 | 130 | 303 | 497 | +4  -9 | +5  -15 | +8  -21 |
| | 4.5 | 0.12 | 0.33 | 0.64 | 124 | 251 | 423 | +13  -13 | +18  -18 | +25  -28 |
| | 8.5 | 0.08 | 0.28 | 0.59 | 93 | 218 | 405 | +18  -16 | +20  -29 | +19  -50 |
| 2070 | 2.6 | 0.17 | 0.41 | 0.79 | 148 | 295 | 521 | +10  -8 | +13  -12 | +11  -14 |
| | 4.5 | 0.06 | 0.28 | 0.61 | 76 | 220 | 420 | +16  -17 | +23  -27 | +28  -56 |
| | 8.5 | 0.03 | 0.13 | 0.33 | 53 | 134 | 270 | +26  -35 | +34  -42 | +38  -67 |
| 2090 | 2.6 | 0.12 | 0.36 | 0.77 | 115 | 249 | 449 | +3  -10 | +11  -22 | +14  -28 |
| | 4.5 | 0.05 | 0.24 | 0.55 | 74 | 196 | 359 | +16  -20 | +25  -31 | +33  -55 |
| | 8.5 | 0.00 | 0.06 | 0.16 | 20 | 85 | 179 | +36  -47 | +45  -68 | +54  -90 |

## 3.2 Projections of multi-RCP annual quantile values

Fifteen-year sliding quantiles for annual indicators of snow and meteorological conditions are displayed in Fig. 3. Figures for each RCP taken separately are available in the Supporting Information (Figs. S2-S4). Values for specific time periods (highlighted in Fig. 3) are provided in Tables 2-3.

Fig. 3 shows the significant interannual variability in snow and meteorologically related indicators in the observations and SAFRAN reanalysis. The observation and reanalysis indicators for snow and meteorological conditions exhibit variations which span the entire range of variations of climate projections, under both historical and early-21[st] century RCPs (the transition between historical and RCP occurs in 2005, which current observations and reanalysis overcross). This indicates that the historical and early-21[st] century RCPs are consistent with the observed range and interannual variability at the considered lo-





**Figure 3.** Quantile values (5%, 17%, 50%, 83% and 95%) over 15-year windows of all GCM/RCM pairs (HIST, RCP2.6, RCP4.5 and RCP8.5), along with annual values of observations (1960-2016) and SAFRAN-Crocus runs (1958-2016) and their respective 15-year running medians (bold full and dotted lines respectively), for: a) $\overline{SD}$, b) $SOD$ and $SMOD$, c) $\overline{T}$, and d) $\overline{P}$. Light grey bars indicate the reference period 1986-2005 and the time slots used in Tables 2-6. **14**





**Table 3.** Quantile values (Q17 = 17%, Q50 = 50%, Q83 = 83%) over 15-year windows, for the reference period 1986-2005 (Ref) in observations (OBS, only Q50), SAFRAN-Crocus (S-C, only Q50) and historical scenario (HIST, *13 GCM/RCM pairs, **4 GCM/RCM pairs corresponding to the ones in RCP2.6), and around the time slots 2030, 2050, 2070 and 2090 for each future scenario (RCP2.6: 4 pairs, RCP4.5 and RCP8.5: 13 pairs), for $STED_5$, $STED_{50}$, $STED_{100}$ (number of days).

| Time slot | | $STED_5$ | | | $STED_{50}$ | | | $STED_{100}$ | | |
|---|---|---|---|---|---|---|---|---|---|---|
| | | Q17 | Q50 | Q83 | Q17 | Q50 | Q83 | Q17 | Q50 | Q83 |
| Ref | OBS | | 135 | | | 96 | | | 39 | |
| | S-C | | 142 | | | 103 | | | 37 | |
| | HIST* | 111 | 136 | 151 | 33 | 90 | 130 | 0 | 27 | 82 |
| | HIST** | 106 | 137 | 153 | 28 | 93 | 134 | 0 | 30 | 81 |
| 2030 | 2.6 | 93 | 121 | 144 | 17 | 63 | 113 | 0 | 11 | 49 |
| | 4.5 | 94 | 123 | 144 | 18 | 62 | 111 | 0 | 13 | 65 |
| | 8.5 | 87 | 119 | 141 | 13 | 65 | 112 | 0 | 12 | 58 |
| 2050 | 2.6 | 83 | 120 | 147 | 13 | 69 | 118 | 0 | 13 | 49 |
| | 4.5 | 77 | 111 | 136 | 4 | 39 | 91 | 0 | 5 | 32 |
| | 8.5 | 55 | 98 | 132 | 0 | 32 | 83 | 0 | 4 | 26 |
| 2070 | 2.6 | 92 | 119 | 143 | 12 | 57 | 110 | 0 | 11 | 60 |
| | 4.5 | 49 | 96 | 129 | 0 | 32 | 86 | 0 | 4 | 29 |
| | 8.5 | 32 | 70 | 104 | 0 | 9 | 47 | 0 | 0 | 4 |
| 2090 | 2.6 | 81 | 109 | 140 | 2 | 49 | 110 | 0 | 8 | 49 |
| | 4.5 | 37 | 91 | 130 | 0 | 25 | 84 | 0 | 3 | 20 |
| | 8.5 | 7 | 35 | 70 | 0 | 2 | 15 | 0 | 0 | 2 |

5 cation, which corroborates the use of the EURO-CORDEX regional climate simulations together with the ADAMONT method and the Crocus snowpack model to address past and future variations of snow conditions in this mountainous area.

For the reference period 1986-2005, the median of annual values of $\overline{SD}$, snow onset date ($SOD$) and snow melt-out date ($SMOD$) is consistent between observations, reanalysis and simulations driven by adjusted historical climate model simulations (HIST using 13 GCM/RCM pairs), with some differences. For example, as can be observed in Table 2, while the

10 $SOD$ median value is similar between observations and simulations (within 1 day), the $SMOD$ median value occurs approximately 10 days later in the reanalysis than in observations, consistent with the 3 cm deviation between the median value of reanalysis-driven and observed $\overline{SD}$. Simulations driven by adjusted historical climate model runs indicate slightly less snow than observations and reanalysis. Similar features can be identified in terms of $STED$ values in Table 3.

In the case where a smaller number of GCM/RCM pairs are considered for the same time period HIST, i.e. when only the 4 GCM/RCM pairs for which RCP2.6 model runs are available and not the 13 GCM/RCM pairs for which RCP4.5 and RCP8.5

5 are available, the indicators calculated for the reference period only taking into account the 4 model pairs available in RCP2.6





(HIST** in Tables 2-3) show very small deviation to the values obtained with 13 GCM/RCM pairs. Quantile values differ by up to 3 cm for $\overline{SD}$ ($\approx$10%), 14 kg m$^{-2}$ for $\widehat{SWE}$ ($\approx$6%) and 3 days for $SOD$ and $SMOD$. For $STED$ quantile values, the largest difference is 5 days ($\approx$15%). This shows that in terms of statistical distributions of annual values of the indicators, the sub-ensemble of four GCM/RCM pairs for which RCP2.6 are available exhibits similar statistical features than the full
10  ensemble of 13 GCM/RCM pairs, in terms of mean trends and spread.

At the scale of 20-year spaced future intervals provided in Tables 2 and 3, all snow-related indicators exhibit a trend towards gradually increased snow scarcity. $\overline{SD}$, $\widehat{SWE}$ quantile values sampled every 20 years generally decrease, $SOD$ increases (later snow onset) and $SMOD$ decreases (earlier snow melt-out date), and $STED$ values decrease. In most cases, climate projections for the 15-year periods centered around 2030 and 2050 depend only slightly if at all on the RCP. The periods centered around 2070 and 2090 show significant deviations between RCPs, with reinforced downwards trends for RCP8.5-based indicators, pursued decrease under RCP4.5 and stabilization or reduced decreasing trend for RCP2.6. In comparison to
the historical model runs during the reference period 1986-2005, not only the median but also the individual quantile Q17 and Q83 values decrease. However the interquantile Q83-Q17 value remains rather constant throughout the century, in comparison with the reference period, except in the late 21$^{st}$ century under RCP8.5 where snow conditions become increasingly ephemeral. For example, in the case of $\overline{SD}$, the Q83-Q17 value of $0.72$ m for the reference period varies for future conditions between $0.62$ m and $0.67$ m for RCP2.6, $0.50$ m and $0.66$ m for RCP4.5 and $0.16$ m and $0.64$ m for RCP8.5 (lowest value at the end of
the century). The variability of snow conditions is therefore projected to remain significant, as large as currently encountered as long as snow conditions remain comparable.

The $\overline{SD}$ quantile-based coefficient of variation (QCV=(Q83-Q17)/Q50) for the reference period is equal to $1.14$, which means that the spread between the Q17 and Q83 quantile values, which comprise 2/3 of the values potentially obtained for a given winter, exceeds the median value itself, highlighting quantitatively how variable snow conditions can be from one winter
to the next. For future conditions, QCV values are never found to be lower than the reference value, and vary between $1.46$ and $1.81$ for RCP2.6, $1.43$ and $2.08$ for RCP4.5, and $1.42$ and $2.67$ for RCP8.5. This indicates that, with the gradual decrease of median and other quantile values for $\overline{SD}$, the interannual/intermodel variability is projected to remain significant and even increase in relative terms (compared to the median value). Very similar results can be obtained when considering $\widehat{SWE}$. In the case of $STED$ values, however, the situation is different especially for $STED_{50}$ and $STED_{100}$ because the number of
snow-scarce winter increase will directly lower the Q83 quantile value while the Q17 quantile value is bounded by $0$ and already equal to this value in the early 21$^{st}$ century for all RCPs for $STED_{100}$ and approaching it by the middle of the 21$^{st}$ century for all RCPs (including RCP2.6) in the case of $STED_{50}$.

### 3.3 Projections of multi-RCP multi-annual mean values

Figure 4 represents the mean $\pm\,\sigma'$ for the same indicators as Fig. 3. Figures for each RCP taken separately are available in
the Supporting Information (Figs. S5-S7). Tables 4-5 also contain values for specific time slots and for additional indicators. Table 6 lists the relative change in $\overline{T}$, $\overline{P}$ and $\overline{R}$ for the same time slots compared to the reference period 1986-2005.



**Figure 4.** Mean ($\mu$) $\pm \sigma'$ of all GCM/RCM combination 15-year running means among each scenario (HIST, RCP2.6, RCP4.5 and RCP8.5), along with 15-year running means of annual values of observations (1960-2016) and outputs of SAFRAN-Crocus runs (1958-2016) at CDP, for: a) $\overline{SD}$, b) $SOD$ and $SMOD$, c) $\overline{T}$, and d) $\overline{P}$. Light grey bars indicate the reference period 1986-2005 and the time slots used in Tables 2-6.





**Table 4.** Values for the mean ($\mu$) $\pm \sigma'$ of 15-year running means, for the reference period 1986-2005 (Ref) in observations (OBS, only $\mu$), SAFRAN-Crocus (S-C, only $\mu$) and historical scenario (HIST, *13 GCM/RCM pairs, **4 GCM/RCM pairs corresponding to the ones in RCP2.6), and around the time slots 2030, 2050, 2070 and 2090 for each future scenario (RCP2.6: 4 pairs, RCP4.5 and RCP8.5: 13 pairs), for $\overline{SD}$, $\widehat{SWE}$ and $SOD$ - $SMOD$ (mm/dd - mm/dd; for RCPs, number of days earlier or later compared to HIST* $SOD$ and $SMOD$).

| Time slot | | $\overline{SD}$ (m) | $\widehat{SWE}$ (kg m$^{-2}$) | $SOD$ - $SMOD$ |
|---|---|---|---|---|
| | | $\mu \pm \sigma'$ | $\mu \pm \sigma'$ | $\mu \pm \sigma'$ |
| Ref | OBS | 0.64 | | 12/09 - 04/16 |
| | S-C | 0.66 | 394 | 12/09 - 04/30 |
| | HIST* | 0.66 $\pm$ 0.09 | 398 $\pm$ 49 | 12/12 $\pm$ 10 - 04/24 $\pm$ 7 |
| | HIST** | 0.66 $\pm$ 0.11 | 400 $\pm$ 48 | 12/10 $\pm$ 11 - 04/23 $\pm$ 8 |
| 2030 | 2.6 | 0.49 $\pm$ 0.16 | 321 $\pm$ 93 | +13 $\pm$ 12   -7 $\pm$ 8 |
| | 4.5 | 0.50 $\pm$ 0.12 | 334 $\pm$ 68 | +11 $\pm$ 9   -11 $\pm$ 10 |
| | 8.5 | 0.48 $\pm$ 0.17 | 312 $\pm$ 78 | +8 $\pm$ 11   -17 $\pm$ 15 |
| 2050 | 2.6 | 0.48 $\pm$ 0.13 | 309 $\pm$ 73 | +6 $\pm$ 11   -17 $\pm$ 14 |
| | 4.5 | 0.40 $\pm$ 0.15 | 279 $\pm$ 72 | +18 $\pm$ 10   -21 $\pm$ 15 |
| | 8.5 | 0.32 $\pm$ 0.09 | 241 $\pm$ 47 | +19 $\pm$ 11   -33 $\pm$ 15 |
| 2070 | 2.6 | 0.47 $\pm$ 0.13 | 325 $\pm$ 69 | +11 $\pm$ 11   -12 $\pm$ 10 |
| | 4.5 | 0.33 $\pm$ 0.13 | 246 $\pm$ 67 | +22 $\pm$ 15   -32 $\pm$ 19 |
| | 8.5 | 0.17 $\pm$ 0.09 | 156 $\pm$ 64 | +32 $\pm$ 10   -46 $\pm$ 17 |
| 2090 | 2.6 | 0.44 $\pm$ 0.07 | 287 $\pm$ 52 | +8 $\pm$ 16   -20 $\pm$ 8 |
| | 4.5 | 0.31 $\pm$ 0.14 | 225 $\pm$ 64 | +24 $\pm$ 20   -34 $\pm$ 12 |
| | 8.5 | 0.09 $\pm$ 0.09 | 101 $\pm$ 74 | +50 $\pm$ 23   -74 $\pm$ 26 |

In contrast to Fig. 3, by design Fig. 4 suppresses most of the effects of the interannual variability, focussing on long-term trends and highlighting the uncertainty components originating from global and regional climate models. As illustrated in Tables 4 and 5, the uncertainty pertaining to multi-annual / multi-model averages is computed based on the standard deviation of the mean of the multi-model multi-annual averages over sliding time periods, as described above. Values for $\sigma'$ ( $= 1.37\,\sigma$) are generally lower for the HIST 1986-2005 period than for the future periods centered on 2030, 2050, 2070 and 2090. For example, $\sigma'$ for $\overline{SD}$ over the HIST 1986-2005 period is equal to 0.09 m, while for all future periods, it is rather on the order of 0.09-0.17 m, except for RCP8.5 towards the end of the century, with $\sigma'$ values 0.09 m, but associated to significantly lower $\mu$ values on the order of 0.09-0.17 m. A similar observation can be made for $\widehat{SWE}$, $SOD$, $SMOD$ and $STED$ values.

In terms of absolute values, as illustrated in Fig. 4, and indicated in Tables 4 and 5, the historical model runs for the reference period 1986-2005 are characterized by about the same amounts of snow on average as in the observations and reanalysis data. This is consistent with the only slight deviation observed between median values in the previous section. As shown in Fig. 4a, the decadal dynamics however differs, with snow conditions (observed and reanalyzed) showing rather stable conditions in the



**Table 5.** Values for the mean ($\mu$) $\pm \sigma'$ of 15-year running means, for the reference period 1986-2005 (Ref) in observations (OBS, only $\mu$), SAFRAN-Crocus (S-C, only $\mu$) and historical scenario (HIST, *13 GCM/RCM pairs, **4 GCM/RCM pairs corresponding to the ones in RCP2.6), and around the time slots 2030, 2050, 2070 and 2090 for each future scenario (RCP2.6: 4 pairs, RCP4.5 and RCP8.5: 13 pairs), for $STED_5$, $STED_{50}$, $STED_{100}$ (number of days).

| Time slot | | $STED_5$ | $STED_{50}$ | $STED_{100}$ |
|---|---|---|---|---|
| | | $\mu \pm \sigma'$ | $\mu \pm \sigma'$ | $\mu \pm \sigma'$ |
| Ref | OBS | 130 | 80 | 32 |
| | S-C | 135 | 89 | 33 |
| | HIST* | $130 \pm 9$ | $84 \pm 13$ | $36 \pm 8$ |
| | HIST** | $130 \pm 9$ | $84 \pm 15$ | $37 \pm 9$ |
| 2030 | 2.6 | $115 \pm 13$ | $62 \pm 22$ | $19 \pm 14$ |
| | 4.5 | $116 \pm 14$ | $63 \pm 15$ | $24 \pm 11$ |
| | 8.5 | $110 \pm 14$ | $60 \pm 22$ | $22 \pm 13$ |
| 2050 | 2.6 | $111 \pm 12$ | $63 \pm 16$ | $18 \pm 8$ |
| | 4.5 | $106 \pm 17$ | $47 \pm 21$ | $14 \pm 10$ |
| | 8.5 | $92 \pm 17$ | $38 \pm 11$ | $11 \pm 6$ |
| 2070 | 2.6 | $115 \pm 10$ | $58 \pm 13$ | $22 \pm 14$ |
| | 4.5 | $92 \pm 23$ | $39 \pm 18$ | $11 \pm 9$ |
| | 8.5 | $67 \pm 21$ | $18 \pm 13$ | $2 \pm 4$ |
| 2090 | 2.6 | $111 \pm 9$ | $56 \pm 6$ | $18 \pm 10$ |
| | 4.5 | $88 \pm 22$ | $38 \pm 21$ | $9 \pm 9$ |
| | 8.5 | $37 \pm 20$ | $6 \pm 9$ | $1 \pm 4$ |

1970s followed by abrupt variation in the mid-1980s, followed by another period of relative stability. Simulations driven by climate model data show a different pattern of $\overline{SD}$ variations, with an earlier reduction in the 1970s, followed by a relative increase in the 1980s followed by another reduction in the 1990s onwards. The length of the observation, reanalysis and historical climate records is too small to generalize, but all three sources of information point towards low frequency variations at the decadal time scale, superimposing on a long-term trend of general snow reduction.

At the scale of 20-year spaced future intervals provided in Tables 4 and 5, similarly to the results of the annual quantiles approach, all snow-related indicators exhibit a trend towards gradually increased snow scarcity. Also similarly, in most cases, climate projections for the 15-year periods centered around 2030 and 2050 depend only slightly if at all on the RCP, the periods centered around 2070 and 2090 show significant deviations between RCPs, with reinforced downwards trends for RCP8.5-based indicators, pursued decrease under RCP4.5 and stabilization or reduced decreasing trend for RCP2.6.

Similarly to the previous section, the values of the indicators are calculated for the reference period either taking into account the 4 model pairs available in RCP2.6 (HIST** ) or the 13 pairs for which RCP4.5 and RCP8.5 are available, see in Tables 4-6.



**Table 6.** Reference values of $\overline{T}$, $\overline{P}$ and $\overline{R}$ for the period 1986-2005 (Ref) from observations (OBS, only $\mu$), SAFRAN (SAF, only $\mu$) and the historical scenario (HIST, *13 GCM/RCM pairs, **4 GCM/RCM pairs corresponding to the ones in RCP2.6). Change ($\mu \pm \sigma'$) in those indicators ($\Delta\overline{T}$, $\rho\overline{P}$ and $\Delta\overline{R}$) for the same time slots and RCPs as in previous tables, compared to the reference period 1986-2005 in HIST*.

| Time slot | Dataset | $\overline{T}$ (°C) | $\overline{P}$(kg m$^{-2}$) | $\overline{R}$ (%) |
|---|---|---|---|---|
| | OBS | 0.9 | 777 | |
| Ref | SAF | 0.9 | 781 | 60.8 |
| | HIST* | 0.4 $\pm$ 0.3 | 762 $\pm$ 54 | 67.4 $\pm$ 3.4 |
| | HIST** | 0.4 $\pm$ 0.4 | 761 $\pm$ 56 | 66.5 $\pm$ 3.9 |
| Time slot | RCP | $\Delta\overline{T}$ (°C) | $\rho\overline{P}$(%) | $\Delta\overline{R}$ (%) |
| | 2.6 | 0.9 $\pm$ 0.3 | 2.5 $\pm$ 6.3 | -7.8 $\pm$ 3.8 |
| 2030 | 4.5 | 1.0 $\pm$ 0.5 | 8.1 $\pm$ 7.3 | -8.4 $\pm$ 3.4 |
| | 8.5 | 1.1 $\pm$ 0.5 | 4.5 $\pm$ 7.5 | -9.3 $\pm$ 4.3 |
| | 2.6 | 1.2 $\pm$ 0.4 | -3.8 $\pm$ 6.9 | -10.0 $\pm$ 5.2 |
| 2050 | 4.5 | 1.6 $\pm$ 0.7 | 5.6 $\pm$ 10.5 | -13.4 $\pm$ 4.2 |
| | 8.5 | 2.1 $\pm$ 0.7 | 5.8 $\pm$ 9.0 | -16.8 $\pm$ 4.2 |
| | 2.6 | 1.2 $\pm$ 0.2 | 1.7 $\pm$ 7.2 | -8.9 $\pm$ 1.3 |
| 2070 | 4.5 | 2.1 $\pm$ 0.8 | 3.3 $\pm$ 9.5 | -18.3 $\pm$ 6.5 |
| | 8.5 | 3.2 $\pm$ 0.9 | 2.7 $\pm$ 13.5 | -27.3 $\pm$ 6.5 |
| | 2.6 | 1.4 $\pm$ 0.5 | -2.6 $\pm$ 9.2 | -12.0 $\pm$ 3.4 |
| 2090 | 4.5 | 2.3 $\pm$ 0.9 | 2.1 $\pm$ 11.8 | -17.7 $\pm$ 8.1 |
| | 8.5 | 4.6 $\pm$ 1.0 | 0.4 $\pm$ 15.5 | -37.3 $\pm$ 7.3 |

Mean values are only slightly impacted for some indicators (e.g. for $\widehat{SWE}$ or $\overline{P}$). This shows that at the interannual time scales, the sub-ensemble of four GCM/RCM pairs for which RCP2.6 are available exhibits similar statistical features than the full ensemble of 13 GCM/RCM pairs, in terms of mean trends and spread.

The $\overline{SD}$ coefficient of variation (CV=$2\times\sigma'/\mu$) for the reference period is equal to 0.27, which illustrates well the suppression of the interannual variability effect. This corresponds to only 24% of the QCV (see above), which indicates that for the reference period and for this case, the interannual variability of annual indicator values plays a stronger role than the inter-model spread for a given year. For future conditions, CV tends to increase, but this is more due to a decrease of $\mu$ in all cases than to $\sigma'$ variations, as shown above. CV remains always smaller than QCV, which indicates that, regardless of the scenario and the time period in the future, the variability/uncertainty related to the inter-model spread (for a given RCP and time period) remains always lower than the inter-annual variations.

Table 6 provides a summary of the meteorological conditions associated to the past and future snow conditions addressed in this study, in terms of multi-annual means. While the mean winter temperature value for the reference period 1986-2005 is on the order of 0.4 - 0.9 °C in the Chartreuse mountain range at 1500 m depending on whether the SAFRAN reanalysis or the historical climate runs are considered, the 15-year period centered on 2030 already exhibits a mean increase of $+1.0 \pm 0.4$ °C





regardless of the RCP. The results for the three RCPs already differentiate for the 2050 lead time, and the difference continues
on widening until the end of the century with $+1.4 \pm 0.5\,^\circ$C for RCP2.6, $+2.3 \pm 0.9\,^\circ$C for RCP4.5 and $+4.6 \pm 1.0\,^\circ$C for
RCP8.5. While the temperature trends are unequivocal, there is no significant trend for total winter precipitation, as shown in
Table 6. The snow/rain precipitation ratio is projected to evolve markedly along with the temperature rise, with a maximum
reduction by $37.3 \pm 7.3\%$ of the snow precipitation share over the total winter precipitation.

### 3.4 Relationship between global temperature trends and local snow and meteorological conditions

Figure 5 shows the relationships between computed variations of the snow and meteorological indicators between 1986-2005
(reference period for this study) and three future time periods (beginning of century (BOC), 2011-2040, middle of century
(MOD) 2041-2070 and end of century (EOC), 2071-2100), and the corresponding global temperature variations simulated by
the driving GCM. This figure uses $\Delta T_{g,EOC-PI}$ as a reference (lower axis). The corresponding relationship to $\Delta T_{g,EOC-Ref}$
is also shown (upper axis), which consists in a shift of $0.62^\circ$C ($\Delta T_{g,Ref-PI}$) although individual $\Delta T_{g,Ref-PI}$ values range
from $0.19$ to $0.84^\circ$C depending on the GCM. Regressions were however computed using the values of $\Delta T_{g,BOC-Ref}$,
$\Delta T_{g,MOC-Ref}$ and $\Delta T_{g,EOC-Ref}$ for each GCM, as well as all three future periods taken together. Table 7 shows the slope
(per global $^\circ$C difference with the Ref value) of the variation of the indicator, as well as the coefficient of determination. With
the notable exception of the cumulated winter precipitation $\overline{P}$, all indicators show consistent relationship with $\Delta T_g$. The slope
of the regression curve is very similar for all three future time periods BOC, MOC and EOC, as well as when all future time
periods are pooled together. The maximum correlation is found for the snow precipitation ratio with a coefficient of determi-
nation of 0.90, followed by local air temperature with a coefficient of determination of 0.86. The worst correlation is found for
$STED_{100}$ ($R^2$=0.48 for all time periods). All snow-related indicators $R^2$ values range between 0.76 and 0.83 (for all future
time periods together), with a trend to lower values for BOC only time period, and higher values for EOC and all time periods
together. The slope of the regression curve, in terms of % variation per global $^\circ$C difference with the Ref value, is larger for
$\overline{SD}$ (about -25%$^\circ$C$^{-1}$ ) than for $\widehat{SWE}$ (-20%$^\circ$C$^{-1}$). Similarly to previous sections, the $SOD$ and $SMOD$ variations are
not symmetrical, i.e. the date of snowpack onset exhibits a lower relative reduction (12 days per global $^\circ$C difference with
the Ref value) than the date of snowpack melt out (17 days per global $^\circ$C difference with the Ref value). Taking the sum of
absolute values of $SOD$ and $SMOD$ as a measure of the variations of total snow season length, it is found that the total snow
season length is decreased by 29 days, i.e. about one month, per global $^\circ$C difference with the Ref value. The slope of the local
temperature regression curve is 1.1 $^\circ$C$^\circ$C$^{-1}$, which indicates that the local rate of warming only slightly exceeds the global
warming rate during the $21^{st}$ century, using this method.

Relating to specific target values of global surface air temperature values variations since the pre-industrial period, Figure 5
and the data provided in Table 8 show for example that for a global temperature increase of 1.5$^\circ$C compared to the pre-industrial
period, the mean variation of mean snow depth at 1500 m altitude in the Chartreuse mountain range is in the order of -25%,
and this value increases very rapidly with increasing global temperature variations, reaching reductions of 65% for 3$^\circ$C global
temperature rise, and even 80% reduction passed 4$^\circ$C temperature rise. However, for a given $\Delta T_{g,EOC-PI}$ value, model runs
spanning several tens of % reduction rate can be sampled (e.g. around 2$^\circ$C), showing that the relationship between global





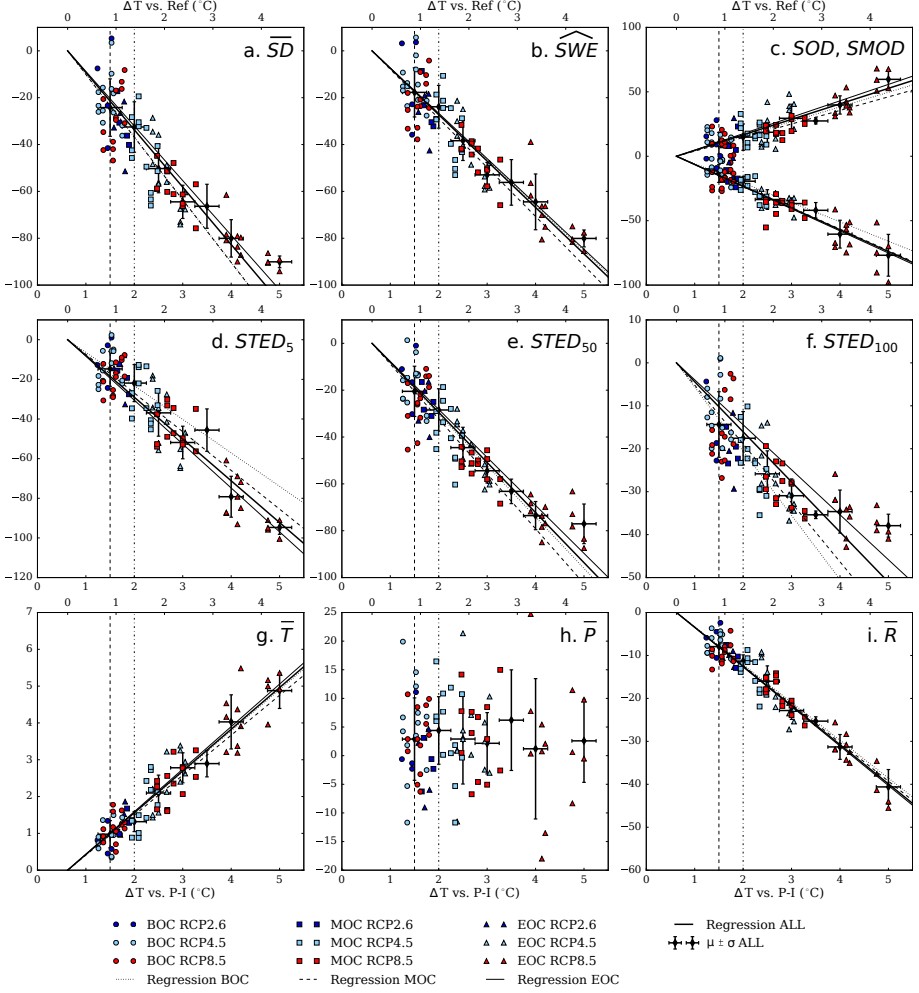

**Figure 5.** Relationship between the variation between end of century (EOC, 2071-2100), middle of century (MOC, 2041-2070) or beginning of century (BOC, 2011-2040) and reference period (Ref, 1986-2005) multi-annual mean of meteorological and snow indicators with the corresponding variation of global mean air temperature, over the reference time period (top axes) and with respect to the pre-industrial period (P-I, 1851-1880) (lower axes). Each point corresponds to a snow or meteorological indicator computed using a given RCP and one GCM/RCM pair, for which the global surface air temperature variation is inferred from the corresponding GCM run: a) $\overline{SD}$ (%), b) $\widehat{SWE}$ (%), c) $SOD$ and $SMOD$ (days), d) $STED_5$ (days), e) $STED_{50}$ (days), f) $STED_{100}$ (days), g) $\overline{T}$ (°C), h) $\overline{P}$ (%), i) $\overline{R}$ (%). 1.5°C and 2°C compared to the pre-industrial are shown with the vertical dashed lines. The regression lines for the variation between EOC, MOC, BOC or all three periods (ALL) and the reference period and $\Delta T_{g,EOC-Ref}$ are indicated (except for $\overline{P}$). Mean values and standard deviations among ALL variations of each indicator for 0.5°C $\Delta T_{g,EOC-PI}$ intervals (± 0.25°C) are displayed as error bars.

temperature values and local impacts is not unequivocal. This is materialized by the standard deviation provided in Table 8. The same applies in terms of trends to all local meteorological and snow indicators (except total precipitation, as noted before).



**Table 7.** Slope ($\alpha$, unit indicated inside brackets) and determination coefficient ($R^2$, no unit) of linear regressions of the variations of indicators between BOC, MOC, EOC or ALL and the reference period (1986-2005) and corresponding global temperature rise since 1986-2005. $\overline{P}$ is not shown. The number of values used for each regression is indicated inside brackets.

| Indicator | BOC (30) | | MOC (30) | | EOC (30) | | ALL (90) | |
|---|---|---|---|---|---|---|---|---|
| | $\alpha$ | $R^2$ | $\alpha$ | $R^2$ | $\alpha$ | $R^2$ | $\alpha$ | $R^2$ |
| $\overline{SD}$ (%) | -26.7 | 0.28 | -26.8 | 0.69 | -23.3 | 0.77 | -24.5 | 0.81 |
| $\widehat{SWE}$ (%) | -19.5 | 0.26 | -20.9 | 0.71 | -19.4 | 0.81 | -19.8 | 0.83 |
| $SOD$ (days) | 11 | 0.02 | 11 | 0.34 | 13 | 0.75 | 12 | 0.72 |
| $SMOD$ (days) | -15 | 0.07 | -17 | 0.46 | -17 | 0.75 | -17 | 0.80 |
| $STED_5$ (days) | -17 | 0.16 | -20 | 0.60 | -22 | 0.88 | -21 | 0.87 |
| $STED_{50}$ (days) | -22 | 0.03 | -23 | 0.54 | -20 | 0.70 | -21 | 0.76 |
| $STED_{100}$ (days) | -15 | 0.22 | -14 | 0.35 | -10 | 0.15 | -12 | 0.48 |
| $\overline{T}$ (°C) | 1.1 | 0.36 | 1.1 | 0.56 | 1.2 | 0.81 | 1.1 | 0.86 |
| $\overline{R}$ (%) | -8.8 | -0.04 | -9.0 | 0.73 | -9.2 | 0.88 | -9.1 | 0.90 |

## 4 Discussion

This study is based on a multi-component ensemble framework in order to provide future values of meteorological and snow conditions at a typical mid-altitude (1500 m) mountain range in the Northern French Alps, accounting for these uncertainty and variability sources in the most consistent and rigorous possible manner. To this end, a multi-component ensemble framework was designed and built, addressing various sources of uncertainty and variability, i.e. several RCPs (RCP 2.6, RCP 4.5 and RCP8.5), feeding several GCM model runs from the CMIP5 intercomparison exercise, which themselves feed various RCM model runs as part of the EURO-CORDEX downscaling exercise, which are adjusted using the ADAMONT method against the meteorological reanalysis product SAFRAN, making it possible to drive a multi-physical version of the energy balance multi-layer snowpack model Crocus. Here we discuss the results obtained for the period from 1950 to 2100, in comparison to reanalysis and comparable observation data for the past period, and with other existing scientific studies for future conditions.

### 4.1 On the comparability between adjusted historical climate model runs and observations and reanalyses

As shown in section 3.1, SAFRAN and Crocus (either multiphysics or default configuration) results show acceptable performance metrics compared to in-situ observations of meteorological conditions and snow conditions, respectively. By definition no performance metrics can be computed on the basis of annual variations between on the one hand observations and/or reanalysis data, and on the other hand adjusted climate model output, because by principle the two are not designed to exhibit synchronous variations. Only multi-annual statistics may be compared, under certain assumptions, which is done in sections 3.2 and 3.3, for the snow indicators defined in this study. Indeed, even over a time scale of 20 years, it is likely and even expected that low frequency variability in the climate, in nature and as it is represented in GCMs, leads to systematic deviations





**Table 8.** Mean value ± standard deviation of the variations of each indicator between ALL and the reference period (1986-2005), for $\Delta T_{g,EOC-PI}$ intervals of 0.5°C. The number of values used in each interval is indicated inside brackets.

| Indicator | 1.5°C (29) | 2.0°C (14) | 2.5°C (21) | 3.0°C (10) | 3.5°C (2) | 4.0°C (8) | 4.5°C (0) | 5.0°C (5) |
|---|---|---|---|---|---|---|---|---|
| $\overline{SD}$ (%) | -24.2 ± 12.3 | -32.5 ± 10.8 | -50.2 ± 10.3 | -64.5 ± 7.1 | -66.4 ± 9.4 | -80.1 ± 8.0 | N-A | -90.1 ± 2.5 |
| $\widehat{SWE}$ (%) | -17.7 ± 10.5 | -23.9 ± 9.1 | -38.4 ± 8.5 | -52.9 ± 5.3 | -56.2 ± 9.7 | -64.5 ± 11.9 | N-A | -80.1 ± 3.6 |
| $SOD$ (days) | 10 ± 8 | 15 ± 10 | 19 ± 8 | 29 ± 11 | 27 ± 2 | 40 ± 7 | N-A | 60 ± 7 |
| $SMOD$ (days) | -13 ± 7 | -19 ± 4 | -34 ± 8 | -37 ± 4 | -42 ± 6 | -60 ± 11 | N-A | -77 ± 16 |
| $STED_5$ (days) | -15 ± 9 | -22 ± 9 | -37 ± 12 | -52 ± 8 | -46 ± 11 | -79 ± 10 | N-A | -95 ± 3 |
| $STED_{50}$ (days) | -21 ± 11 | -28 ± 9 | -45 ± 9 | -54 ± 5 | -63 ± 5 | -74 ± 6 | N-A | -77 ± 8 |
| $STED_{100}$ (days) | -14 ± 8 | -18 ± 6 | -26 ± 5 | -31 ± 4 | -35 ± 1 | -35 ± 5 | N-A | -38 ± 3 |
| $\overline{T}$ (°C) | 1.0 ± 0.4 | 1.3 ± 0.3 | 2.1 ± 0.5 | 2.8 ± 0.4 | 2.9 ± 0.4 | 4.0 ± 0.7 | N-A | 4.9 ± 0.5 |
| $\overline{P}$ (%) | 2.9 ± 7.2 | 4.4 ± 5.9 | 2.9 ± 7.9 | 2.2 ± 5.4 | 6.2 ± 8.8 | 1.2 ± 12.3 | N-A | 2.6 ± 7.2 |
| $\overline{R}$ (%) | -7.9 ± 2.6 | -11.3 ± 1.4 | -16.0 ± 3.6 | -22.8 ± 2.1 | -25.3 ± 1.0 | -31.3 ± 2.9 | N-A | -40.6 ± 4.0 |





at this time scale, which the statistical adjustment method can only partially mitigate. For the reference period 1986-2005, the match between observation and reanalysis data, and historical GCM/RCM runs is nevertheless satisfying. However, it is also clear from Fig. 3 and Fig. 4 that the match is not as good for a period extending back into the past, with a tendency for adjusted climate model data to provide reduced snow conditions compared to observed and reanalyzed data for the period before 1985. While the reasons for such a behaviour are likely multiple, it is certainly influenced by the fact that this period is almost inde-
pendent from the time period used for the adjustment over which the ADAMONT method was applied (1980-2011), and during
which major climate shifts occurred (Reid et al., 2015). This could also be due to the fact that Crocus model outputs result from the interaction between various meteorological variables, both in terms of mean values but also their day to day variations, especially precipitation and temperature conditions which together yield either to rain or snow precipitation. By design, the ADAMONT method adjusts the variables independently from each other (Verfaillie et al., 2017). Even if special care is taken to minimize the disadvantages of this approach, such as the use of weather regimes for the quantile mapping statistical adjustment
method, or the final quantile mapping applied to rain and snow precipitation separately to mitigate temperature/precipitation detrimental interactions (Verfaillie et al., 2017), some interaction terms probably remain uncorrected. The adjustment method also probably exerts an influence on the variability during the historical period, which may be responsible for the overall lower spread (either expressed in terms of quantile-based coefficient of variation of annual values or the coefficient of variation of the interannual means) compared to future projections. Indeed, by design the adjustment method attempts to bring reanalysis
meteorological data and historical model runs to the same ground in terms of quantile distributions, which inevitably reduces the spread of interannual variations between different models. This is visible in the analyzed results, because the reference time period used 1986-2005 is included in the period used for the statistical adjustment method. In addition, the lower spread, compared to future periods of 15 years, could also be due to the fact that the reference period is longer than the future time periods considered, so that a wider range of climate conditions are sampled in the multi-annual mean, thereby bringing closer
the values originating from the various RCMs.

## 4.2 Uncertainty and variability sources

The study uses multi-component ensembles to address uncertainty and variability sources, which are analyzed through indi-
cators computed using various sub-ensembles. Based on the results shown above, it clearly appears that snowpack modeling errors, due to uncertain physical knowledge of processes at play and their imperfect implementation in the model, can be re-
25 sponsible for a significant fraction of the uncertainty pertaining to future climate projections, consistent with previous results obtained based on observations at instrumented sites (Essery et al., 2013; Lafaysse et al., 2017). While this must be taken into account for a fully comprehensive assessment, evidence from this study suggests that under the conditions of the Northern French Alps the uncertainty component attributed to the snowpack modeling errors alone is on the order of 20%, which is significant but of second order compared to the spread originating from multiple climate models.
Initially motivated by the fact that the number of GCM/RCM model pairs was different for RCP2.6 (4) and RCP4.5 and RCP8.5 (13), which could lead to different multi-model statistics and the impossibility to compare them specifically, we compared the statistics for indicators based on either 4 or the full ensemble of 13 GCM/RCM pairs for the historical period.



Both in terms of statistics distributions of annual values for a period of 20 years (1986-2005) or in terms of multi-model spread of multi-annual average values, results were extremely close for the full and sub-ensemble. While it remains desirable, when possible, to use the largest possible number of different GCM/RCM pairs in order to mitigate the impact of multi-model variability and climate internal variability, this tends to show that, in this case, robust results can be obtained using a subset of

a few models dealt with appropriately. However, as shown in Figure 5, individual GCM/RCM pairs only sample imperfectly the range of possible future climate conditions, so that choosing, randomly or not, a too small number of GCM/RCM pairs, would inevitably lead to biased results. This is consistent with the fact that the variability of snow conditions is primarily dominated by interannual variability, over which inter-model spread superimposes an additional uncertainty component. It is very likely that the 4 GCM/RCM pairs used in this study, which feature RCP2.6, RCP4.5 and RCP8.5 model results, possess

appropriate interannual variability properties and overall no major deviation from the average behaviour of the full ensemble of 13 GCM/RCM pairs, which leads to the fact that similar statistics are found with these 4 model pairs than for the 13 full ensemble. It is not certain that a similar result would be obtained by picking randomly 4 GCM/RCM pairs within the full ensemble available (see Figure 5 for contrasted individual model behaviour).

### 4.3    General trends and added value of the approach developed

That natural snow conditions at 1500 m in the Northern French Alps are projected to decrease under ongoing climate change is an expected result, which deserves however to be put in perspective with other existing studies on the matter. Figures 3-4 and Tables 2 and 4 indicate a general decreasing trend in $\overline{SD}$ towards the end of the century ($\approx -0.8$ cm per decade for RCP2.6, $-3.2$ cm per decade for RCP4.5 and $-6.5$ cm per decade for RCP8.5 over the period 2030-2090), accompanied by a shortening of the snow season (later $SOD$ and earlier $SMOD$). This is consistent with previous results from Steger et al. (2013) for the

1000 - 1500 m a.s.l. range in the European Alps. The magnitude of the $\overline{SD}$ decrease is similar to the one found by Marty et al. (2017a) for the Aare and Grisons regions in Switzerland, although their GCM/RCM models and future scenarios differ from ours. This trend is visible for all scenarios, but stronger for RCP8.5. At the end of the century, simulations carried out under this scenario predict an increasingly ephemeral snow cover (multi-annual mean value of $9 \pm 9$ cm for the 2090 time slot, see Table 4) and more frequent snow-dry seasons (Figs. 3-4 and Tables 2 and 4). The shortening of the snow season is projected to

become asymetric towards the end of the century, with a stronger reduction in spring than in autumn (Tables 2 and 4), similar to findings from Steger et al. (2013) and Marty et al. (2017a). The decreasing $\overline{SD}$ trend is also combined with a decrease with time of $\widehat{SWE}$ ($\approx$ -6 kg m$^{-2}$ par decade for RCP2.6, -18 kg m$^{-2}$ per decade for RCP4.5 and -35 kg m$^{-2}$ per decade for RCP8.5 over the period 2030-2090, Table 4) and of $STED_5$ (as in Marty et al. (2017a)), $STED_{50}$ and $STED_{100}$ (Table 5).

     Figures 3-4 also indicate a strong increasing trend in $\overline{T}$ for the $21^{st}$ century ($\approx +0.08°$C decade$^{-1}$ for RCP2.6, $+0.22°$C

decade$^{-1}$ for RCP4.5 and $+0.58°$C decade$^{-1}$ for RCP8.5 over the period 2030-2090), but no significant trend in $\overline{P}$. Compared to the reference period 1986-2005, $\overline{T}$ increases by $1.4 \pm 0.5°$C in 2090 for scenario RCP2.6, $2.3 \pm 0.9°$C for scenario RCP4.5 and $4.6 \pm 1.0°$C for scenario RCP8.5 (Table 6). Values for the change in $\overline{T}$ and $\overline{P}$ are comparable to Steger et al. (2013) and Marty et al. (2017a), even though their GCM/RCM models and future scenarios differ from ours. The insignificant trend in $\overline{P}$ and its variable sign depending on the projections is fully consistent with previous studies identifying the internal variability





of climate as the main uncertainty component for precipitation in the Alpine region all along the $21^{st}$ century (Lafaysse et al.,

2014; Fatichi et al., 2014). Table 6 further shows a strong decrease in $\overline{R}$ (by 2090, -12.0 ± 3.4% for RCP2.6, -17.7 ± 8.1% for
RCP4.5 and -37.3 ± 7.3% for RCP8.5, compared to 1986-2005), with values very similar to Frei et al. (2017).

Beyond the general trends, which provide an unsurprising -yet required- update of previous assessments based on older
climate scenarios applied to the French Alps (e.g Rousselot et al., 2012; Castebrunet et al., 2014; Piazza et al., 2014), the main
added value of the approach developed here lies in its ability to capture high-order moments of possible snow futures. For

example, that the year-to-year variability of snow conditions on the ground remains as large as currently, and even increases in
relative terms, may be of equal, if not higher significance, to stakeholders operating in the alpine environment, than the long
term trends. Such results can only be attained making use of a sufficiently large number of independent global and regional
climate models, the EURO-CORDEX database corresponding to a significant achievement of the climate modeling community
enabling such impact studies to take place.

**4.4  Link with global temperature increase**

The international framework for climate negotiations, culminating at the yearly Conferences Of Parties (COP), and basing the
technical part of its decision process on IPCC assessments, shows a strong tendency to focus on global temperature changes.
While for a number of reason this approach is limited and only partially represents climate change (Rogelj et al., 2015; Millar
et al., 2017; James et al., 2017), its infusion in the public debate at all levels, from the international, national and even local level,

makes it relevant to discuss and illustrate local impacts of global climate change. With Figure 5 and Tables 7 and 8 we provide
such a link, thereby highlighting the specific sensitivity of the mountain meteorological and snow conditions to global climate
conditions. Such figures allow stakeholders interested in snow and meteorological conditions at the local scale to directly infer
the consequences of climate policies in their socio-economic domain (James et al., 2017; Marty et al., 2017a). However, using
only such an approach with a focus on the end of the $21^{st}$ century, may lower the impact of the results and the motivation

of stakeholders, if the consequences appear too distant in time. The power of the approach shown in this article, is that, not
only it makes it possible to infer EOC impacts of climate change, but also provides a continuous vision of past and current
climate context, and its most likely evolution according to state-of-the-art GCM/RCM pairs driven by RCPs. Furthermore, the
data obtained indicate that the response of local meteorological and snow conditions is essentially the same regardless whether
data from the beginning or end of the century are sampled. This indicates that the seasonal snowpack responds in a reversible

way to global-scale climate change, and the near-term and mid-term responses can be used, in addition to the end of century
information, to infer the relationship between local and global conditions using a larger dataset thereby providing more robust
assessments of the influence of the global air temperature on local snow and meteorological data. This is all the more relevant
that none of the GCM used for this study predict EOC warming below 1.5°C compared to pre-industrial levels, so that using
less distant future time periods makes it possible to assess the response of the local snow conditions to 1.5°C and 2°C difference
in a more robust way than EOC only (see Table 8) (James et al., 2017). Even for the lowest level of global warming, none of
the model results predict that local snow conditions will be unaffected by climate change, the minimum level of decrease of
mean winter snow depth being on the order of 25% for 1.5°C global increase since pre-industrial period.





In more details, these results highlight several discussion points. First of all, it is remarkable that the regression line of the local mean winter temperature with global temperature increase shows a slope of 1.1 $°C°C^{-1}$, which represents a low additional warming of the mountain environment in contrast to previous studies (Durand et al., 2009a; Pepin et al., 2015), although elevation dependent warming is generally maximal in the fall and springtime, while our target period covers mostly wintertime. This low elevation dependent warming enhancement factor could be due to the fact that the RCM grid points used for our analysis are at lower altitudes, from 612 to 1085 m, with a mean value across all RCMs of 880 m. Snow conditions at such altitude levels are generally limited already at present time, so that the local snow albedo feedback which drives much the elevation warming (Pepin et al., 2015) may be limited at such a low elevation. Addressing this issue in more detail is left open for future research, as it may imply that the temperature trends identified in this study are underestimated for this reason. Second, it is interesting to note that the relationship between snow conditions and global air temperature is different for winter mean snow depth and peak SWE. The latter shows a lower sensitivity ($-20\%°C^{-1}$) than mean snow depth ($-25\%°C^{-1}$), see Table 7. While this is first due to the different nature of the indicators (peak SWE value vs. mean winter snow depth value), this may also be due to the fact that rain on snow events (whose frequency is projected to increase) can positively contribute to SWE, through refreezing of the precipitation water in the snowpack, while not contributing to increasing snow depth. This shows that the difference of response of the snow-related indicators must be carefully assessed depending on the target socio-economic domain, because specific snow-related variables may provide distinct messages regarding their impact. While global temperature is well correlated to the snow indicators, the slope of the regression curve is not the same for all indicators, illustrating the usefulness of using a detailed snowpack model to predict the impact of climate variations of snow conditions, accounting for a maximum amount of processes operating at the boundaries and within the snowpack.

## 5 Conclusions

This study introduced a multi-component ensemble framework in order to provide future values of meteorological and snow conditions in mountainous regions, exemplified for a typical mid-altitude (1500 m) mountain range in the Northern French Alps. The multi-component ensemble framework makes it possible to account for the various sources of uncertainty and variability, which affect future climate projections, and some if not most of whom are often neglected in past and still ongoing climate change impact studies. The multi-ensemble framework developed here addresses various sources of uncertainty and variability, drawing on several RCPs (RCP 2.6, RCP 4.5 and RCP8.5), feeding several GCM model runs from the CMIP5 intercomparison exercise, which themselves feed various RCP model runs from the EURO-CORDEX downscaling exercise, which are adjusted using the refined quantile mapping method ADAMONT against the meteorological reanalysis SAFRAN, making it possible to drive a multi-physical version of the energy balance multi-layer snowpack model Crocus. The primary material the method draws on is a series of snow and meteorological indicators defined at the annual scale, representing various features of the winter season (mean annual snow depth, peak Snow Water Equivalent, date of inception and melt out of the snowpack, mean air temperature, cumulated winter precipitation etc.), which are computed from daily values of the variables representing meteorological and snow conditions (here temperature, precipitation, snow depth and SWE).





Based on an analysis of various sub-ensembles of past, current and future observations and simulations, spanning the period from 1950 to 2100, and focussing on this particular yet representative geographical setting, the main conclusions of this study are that:

– Uncertainty arising from physical modeling of snow can account to 20% typically of the simulation results, although the multiphysics is likely to have a much smaller impact on trends, because of the systematic nature of a large fraction of the error sources considered.

– The ADAMONT method appropriately adjusts the output of the EURO-CORDEX GCM/RCM model runs, making it possible to drive an energy balance land surface model such as Crocus based on the chronology of the driving climate model, thereby leveraging the caveats of using delta change methods applied to past observations, which do not make it possible to take into account variations in seasonality or climatically-variable weather patterns (blocking, extreme precipitation events, etc.). The method can be readily applied to the next generation of climate model runs, generated using refined greenhouse gas emission scenarios and/or improved model components (Rogelj et al., 2015; Millar et al., 2017). This should make it possible to update quicker than in the past the climate change impact assessment, reducing the phase lag between the production of assessments of global, regional and local climate variations and of their impacts.

– The four GCM/RCM models within the EURO-CORDEX ensemble, which provided not only RCP4.5 and RCP8.5, but also RCP2.6 model runs, exhibit similar statistics at the interannual and multi-annual scale than the 13 full ensemble, making results obtained for RCP2.6 comparable with results obtained for RCP4.5 and RCP8.5 even though they are not based on the same number of models. This result may not generalize to any sub-ensemble of the available GCM/RCM runs of EURO-CORDEX, therefore we consider preferable to use as many as possible GCM/RCM model runs in ensemble-based assessments.

– Ensembles of climate projections generated under RCP2.6, RCP4.5 and RCP8.5 are rather similar until the middle of the $21^{st}$ century, with the continuation of the ongoing reduction in mean interannual snow conditions, but maintained interannual variability of snow conditions, playing even an increasing relative role along with the decrease of mean snow conditions. The interannual variability of meteorological and snow conditions generally induces a stronger spread in potential annual values than the inter-model dispersion (for a given RCP).

– The impact of the RCP becomes significant for the second half of the $21^{st}$ century, with overall stable conditions under the RCP2.6 scenario, and continued degradation of snow conditions along with increased air temperature variations for RCP4.5 and 8.5, the latter leading to frequent occurrence of ephemeral if not almost potentially completely snow-free snow conditions at the end of the century.

– Variations of local meteorological and snow conditions show significant correlation with global temperature levels (using 30 year means), with respect to pre-industrial levels. For example, the mean variation of mean snow depth at 1500 m altitude in the Chartreuse mountain range is in the order of -25% for 1.5°C global temperature rise with respect to pre-





industrial levels, and this value decreases very rapidly with increasing global temperature variations, reaching reductions
of 65% for 3°C global temperature rise, and even 80% reduction beyond 4°C temperature rise.

While this work provides scientific results directly exploitable for snow and meteorological conditions at 1500 m altitude in the Chartreuse mountain range, our results do not directly allow extrapolation of the conclusions in other mountain regions in France or other elevations. Based on the methodological framework introduced here, this will be tackled in subsequent exhaustive investigations, based on the entire wealth of data available in the SAFRAN reanalysis for the French Alps and Pyrenees (Durand et al., 2009b, a; Maris et al., 2009). The method can obviously be applied beyond French borders, provided that an adequate long-term observational dataset can be used as a basis for RCM output adjustment using the ADAMONT method (Verfaillie et al., 2017).

Beyond the geographical scope, which can be extended to address a wider diversity of territorial climate-related challenges, sector-specific further applications can now be considered. For example, the adjusted climate scenarios can be projected on sloping surfaces, making it possible to compute Crocus snowpack model runs able to tackle avalanche hazard evolution, thereby upgrading and consolidating the results of Castebrunet et al. (2014). Also, the adjusted climate scenarios could be employed to simulate snow conditions on ski slopes in French ski resorts, drawing on the method developed by François et al. (2014) to be applied using the version of Crocus accounting explicitly for snowmaking and grooming (Spandre et al., 2016b). This method has shown significant potential to account simultaneously for the impact of natural snow precipitation and temperature conditions (driving the capability to produce snow) on the operating capabilities of alpine ski resorts over the past decades (Spandre et al., Under Review). It is now ready to be applied for future conditions, drawing on the framework developed in this study. Conversely, it is re-emphasized here that, while variations in natural snow conditions as projected in this work are likely to affect operating conditions of ski resorts, no quantitative conclusions can be made, given that snow management practices induce significant changes in operating conditions, which depend on intricated factors related to temperature and precipitation (Hanzer et al., 2014; Spandre et al., 2016b). Beyond these seasonal snow applications, the method is ready to use for hydropower potential, water resources assessments, glacier mass balance studies, ecology, natural hazards related to meteorological conditions and more generally environmental impact studies which can be based on mechanistic derivations of the impact of meteorological conditions on the socio-ecosystemic compartment under investigation.

*Competing interests.* TEXT

The authors declare no competing interests.

*Acknowledgements.* We thank CEN staff members for contributing to observational data acquisition at the Col de Porte site (M. Dumont, B. Lesaffre, P. Lapalus, J.-M. Panel, D. Poncet, P. David and M. Sudul) and the SAFRAN reanalysis (Y. Durand, G. Giraud and L. Mérindol). Fruitful discussions with colleagues at Météo-France (CNRM, S. Somot and S. Planton; DCSC, J.-M. Soubeyroux) have been beneficial for the present study, too. This study benefited from funding from the French Ministry for Ecology (MTES) through the GICC program and





15   ONERC, in the framework of the ADAMONT project, from the Interreg project POCTEFA/Clim'Py and from the IDEX Univ. Grenoble

Alpes Cross Disciplinary Project "Trajectories". CNRM/CEN and Irstea are part of LabEX OSUG@2020 (ANR10 LABX56).





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
