# Peer review of "Multi-component ensembles of future meteorological and natural snow conditions in the Northern French Alps (Chartreuse, 1500 m altitude)"

_The Cryosphere, 2017_

## Referee Comment (RC1) · Anonymous Referee #1 · 3 Jan 2018

The work by Verfaillie et al. presents a comprehensive analysis of past and future snow conditions at a mid-elevation mountain range in the French Alps. The regional SAFRAN re-analysis and bias-adjusted RCM experiments covering three greenhouse gas emission scenarios are used to drive the Crocus snowpack model, and model simulations are compared against observations at a single measurement site. The snowpack model is employed in a multiphysics ensemble approach which allows for an assessment of the contribution of snowpack modelling uncertainty to the overall projection uncertainty. Results for a range of snow indicators are presented. Concerning the overall future degradation of the snowpack they largely confirm previous works, but also provide a number of new and useful insights that are at least valid for this specific

case.

Overall, I consider the paper as a relevant and interesting piece of work. The methods and data used are comprehensively described and are well introduced (except for the downscaling and bias-adjustment method ADAMONT, which is however explained in detail in a previous paper). The methodological approach is sound and valid. The introduction and the discussion properly refer to existing works in this field, and the conclusions are well based on the results obtained. There are no language issues, and the topic clearly fits into the journal's scope. As such, I could generally recommend a publication of the work. However, a few minor and one major issues remain, and I'd suggest to ask the authors for a revision of their work in these respects before final publication. Minor issues are listed at the end of this review.

The remaining problem with the existing manuscript is its rather technical touch and the wealth of information that is presented in terms of data sets, emission scenarios, scenario periods, methods and especially snow indicators. The comprehensiveness of the work is impressive, but the reader very easily gets lost in this large amount of information that is presented in the text, in the tables and in the figures. These information might be very useful for local stakeholders operating in this very region and being affected by snow conditions, but for a truly useful contribution to the scientific community the results and their presentation need to be much better streamlined in my opinion. The generally most interesting part of the work is probably the entire methodological approach and the possibilities that arise from it. The very detailed results for a representative elevation of 1500 m in the Chartreuse mountain range are more of a case study, and the details of their presentation should receive less emphasis. One option might be to remove parts of the analysis entirely from the manuscript and place it in an additional, accompanying publication (e.g. the multiphysics ensemble analysis which is only briefly described in the results and which has much more potential to be analyzed in more detail). Another option is to move some of the material from the main manuscript to the supplement. This could for instance concern several of the snow

indicators (like onset and meltout date), that are in any case only briefly discussed. In combination with such a streamlining, I'd suggest to put a little more emphasis on the actual processes that are responsible for the identified future changes in the snow indicators. Little is so far said about that. The Crocus model output surely provides an opportunity to do so (e.g., separate analysis of snow accumulation and snow melt amounts). In this respect, the relation of the snow indictaor changes to the GLOBAL temperature change is not very helpful and the authors should think about putting the LOCAL temperature change into focus (though I completely understand the choice of the global scale given political climate targets). Such a shift of the focus away from details of the case study and towards a more methodological and process-oriented analysis would be very worthwhile in my opinion. Apart from this and as said before, I consider the manuscript as being of high quality and of general relevance for the readership of the journal.

With kind regards.

Minor issues ===== Spatial scale of the Crocus application: What remains somehow unclear is the spatial setup of the Crocus model. I assume the authors use a single-site setup, driven by the outcomes of the SAFRAN reanalysis and of the ADAMONT downscaling method for a representative 1500 m elevation range in the Chartreuse massif. Is that the case? If so, this needs to be clearly said and described in some more detail. It would imply that the results shown are only valid for that specific elevation range in this massif. What about other elevations then? Is it possible to come up with some speculation here as well? Snow projections will surely strongly depend on the elevation considered, and some placement of the results into a broader spatial context would be helpful.

Page 1 Line 2: "investigates" instead of "introduces" is probabaly the better choice.

Page 1 Line 9: "reduction in mean interannual snow conditions" is rather unclear.

Page 2 Line 30: "they" instead of "there".

Page 3 Line 22: The term "currently" is probably wrong. At this point, more GCM-RCM chains are available from EURO-CORDEX. The authors just either specify their date of access of the data base or justify their selection of all available model chains.

Page 4 Lines 8-15: Please check: Is SAFRAN really ONLY available over mountain ranges? To my knowledge, entire France is covered.

Page 8 Lines 5-21: This method description is rather confusing and very hard to follow. Please streamline. Figure 1: The STEDx should represent some duration of exceedance and hence need to be represented by some horizontal range in this graph. The representation by single vertical arrows is probably wrong, please check.

Page 9, line 1: Temperature changes are surely not computed in a relative manner, please check.

Page 19 Lines 6-10: I assume this is simply an effect of random internal variability at decadal scale, could that be (simulations out-of-phase with reality)? Please clarify.

Page 23 Line 31: Isn't it rather random variations (instead of systematic variations)?

Page 25 Lines 14-16: Is this really the case? Why should a matching of quantile distributions reduce interannual variations? please check and better explain.

---

## Referee Comment (RC2) · Anonymous Referee #2 · 15 Jan 2018

This paper analyses the mean state and variability in historical and future snow conditions at a mid-elevation location in the French Alps. Analysis is based on output from a physical snowpack model (Crocus) driven by a regional historical reanalysis (SAFRAN) and a suite of historical and future RCM simulations. In situ observations from a nearby location are also used for comparison of the historical conditions.

The methodology used in the paper is novel and represents an improvement on previous studies. Results are generally well described compared to existing literature. However I believe there are several changes required and details to clarify before I can recommend final publication.

[Figure]

Specific Comments:

Table 1 obscures the fact that there are only 5 distinct GCMs simulations that sample natural variability. I think this should be pointed out explicitly.

Figure 2a-c: In the supplementary material for the RCP8.5 scenario there is a distinct change in the stddev as average snow depth becomes small. It's therefore unfair therefore to suggest that the stddev is stationary based on the RCP4.5 scenario. I think it would be fairer to show the RCP8.5 scenario in the paper and place RCP2.6 and 4.5 in the SI.

Figure 2d: The large relative contribution to combined model uncertainty arising from snowpack model multiphysics compared to RCM/GCM inter-model variability seems difficult to reconcile with plots 2a-c (I realize there is interannual variability in Figs 2a-c, while much of this signal is removed in the multi-annual average). Is it possible to present 15 year running means for the 13 RCM/GCM tracks shown in Fig 2c or at least for a representative subset of the 13 pairs along with Fig d (either in this same figure, or in a separate figure similar to how you subsequently separate quantile plots based on annual frequency data and multi-annually averaged data)? This may allow the reader to get a better sense of the relative spread in the ESCROC ensembles compared to the RCM/GCM inter-model variability. Further to this point — in the text there are several times that you refer to the relative fraction of uncertainty contributed by snowpack model multiphysics as ~20%, yet this graph shows that is can be as high as 80%. Please justify the use of 20%. Is there a rationale for why the uncertainty due to snowpack model multiphysics is higher in the historical period, even though the combined model uncertainty remains fairly constant?

Figure 3 and 4: The quantiles from different RCPs overlap too much in these figures to discern one set of shading from the others. I suggest showing RCP8.5 only in these figures. The results from the other RCP scenarios are provided in tabular form in the main document which I think is sufficient (along with the plots of other RCPs in the SI).

Figure 3: Why do the SOD and SMOD for RCP8.5 begin to encompass summer months? Is this an error with the calculation?

P8.L15-21: Please clarify why you use a distance of +/- 1.37*sigma from the mean for the 17th and 83rd percentiles? Shouldn't the 17th and 83rd percentiles be 0.95 sigma away from the mean such that CV=1.9*sigma/mu?

P25.L28-29: Please rephrase in order to account for the relative component attributed to snowpack modeling errors in both future and historical periods.

P27.L10: This statement depends on the RCP scenario. It is not the case for RCP8.5.

P28.L15-17: Could there also be a change in the mean density of snowfall occurring at the location?

P29.L8-10: Please reword or justify the claim that snowpack modeling uncertainty is typically 20% when Figure 2d shows it can be up to 80%. I agree that it may have a smaller impact on trends.

P29.L11-17: The ADAMONT method was evaluated in a previous paper. Please be clear as to which aspects of these conclusions were accomplished in this paper. Further to this point, while you argue that your methodology is an improvement to previous studies that use delta change methods, your assessment of the results says that they, in fact, agree with these previous studies. Under what circumstances might you expect to see differences? Is it possible to provide a direct comparison between your methodology and a delta change method for this location or to highlight statistics that would differ between the two methods?

Technical corrections/Suggestions:

I find the use the phrase "annual-scale" a bit unnatural. I suggest using "annual indicators" as you occasionally do (P8.L15) throughout the paper.

Similarly, there are places in the paper where you might consider replacing the word

"variation" with "change", "response", "difference" or "variability", but I'm having trouble articulating a clear rule to follow in this. Variation is frequently reserved to specify a very small adjustment.

Title: "Multi-component ensembles. . .."

P1.L2: "This article investigates the climatic response of a series of indicators for characterizing annual snow conditions and corresponding meteorological drivers at 1500 m altitude in the Chartreuse mountain range in the Northern French Alps. "

P2.l30: "because they are newer. . .."

P3.L2-4. Please rephrase these two sentences to make the typical delta-change approach clearer.

P8.L15: "Moments of multi-year averages: A running average of annual indicator values is computed (typically with a 15 year sample window), for a given RCP and for each GCM/RCM pair."

P9.L1 "for 15-year windows around each future time period t and each RCP r" P9.L3: "i.e." in place of "e.g."

P10.L19: "It highlights the significant interannual variability in observed, reanalyzed and climate model datasets."

P11.L37: "which highlights the need for appropriate data synthesis methods". Please elaborate.

P21.L10: to widen

P23.L27: "By definition no performance metrics pertaining to annual variations can be computed between the adjusted climate output and either observations or reanalysis data, because the two are not designed to exhibit synchronous variations."

P25.L4: "independent from the time period for calibration of the ADAMONT adjustment

method (1980-2011)…."

P25.L9-10: "or applying the final quantile mapping separately to rain and snow precipitation in order to mitigate detrimental interactions between temperature and precipitation (Verfaillie et al., 2017)…."

P25.L30: "Because the number of GCM/RCM model pairs was different for RCP2.6 (4) and RCP4.5 and RCP8.5 (13), we compared the statistics for indicators during the historical period based on the 4 RCP2.6 pairs alone, as well as the full ensemble of 13 GCM/RCM pairs."

P26.L11: "similar statistics are found for these 4 model pairs as for the full ensemble of thirteen."

P26.L24: I'm not sure what you mean by "snow-dry" seasons. Seasons without snow on ground or without snowfall occurring at this location at all?

P26:L26-28: "The decreasing SD trend is also combined with a decreasing SWE trend ($\sim$ -6 kg m$-2$ per decade for RCP2.6, -18 kg m$-2$ per decade for RCP4.5 and -35 kg m$-2$ per decade for RCP8.5 over the period 2030-2090, Table 4) and a decreasing trend in duration of STED5 (as in Marty et al. (2017a)), STED50 and STED100 (Table 5)."

P27.L33: "This is all the more relevant in that none of the GCMs used for this study…."

P28.L6-8: "…in contrast to previous studies (Durand et al., 2009a; Pepin et al., 2015). This result may stem in part from the fact that although elevation dependent warming is generally maximal in the fall and springtime, our target period covers mostly wintertime. Alternatively, this low enhancement factor could be due…."

P28.L26-28: "The multi-component ensemble framework makes it possible to account for the various sources of uncertainty and variability that affect future climate projections, some of which are neglected in both previous and ongoing climate change impact studies."

P28.L28-32: Split into more than 1 sentence.

P28.L32-36: "The method defines a series of annual snow and meteorological indicators that represent various aspects of winter snow conditions...."

P29.L21: "exhibit similar statistics at the interannual and multi-annual scale as the full 13-member ensemble, ...."

P29.L26: "but maintained interannual variability...... " Rephrase.

P29.L28-29: "As assessed in this study, for this location, interannual variability is larger than inter-model spread for a given RCP scenario."

P29.L32-33: "the latter leading to frequent occurrence of ephemeral or nearly snow-free conditions at the end of the century."

P29.L25: "For example, the change in mean snow depth"

P30.L4: "this value changes very rapidly": I dislike the wording that it changes "rapidly" since the changes on Figure 5 are quite linear (except for STED100). Please rephrase. I suggest something along the lines of "the magnitude consistently increases along with global mean temperature reaching reductions of 80% beyond 4°C of global warming.

P30.L8-10: "These locations may be investigated in the future, based on the methodological framework introduced here and the data available in the SAFRAN reanalysis for the French Alps and Pyrenees (Durand et al., 2009b, a; Maris et al., 2009)."

Figure 2 Caption: "c) Ensemble of Crocus model configurations driven by the 13 RCP4.5 GCM/RCM pairs; each GCM/RCM pair is displayed with a different color."

Figure 4 Caption: "Ensemble spread in 15-year running mean ( $\mu \pm \sigma$' ) of all GCM/RCM pairs for each scenario (HIST, RCP2.6, RCP4.5 and RCP8.5), along with 15-year running means of observations (1960-2016) and SAFRAN-Crocus runs (1958-2016) at CDP, for: ...."

Figure 5 Caption: "Response of local meteorological and snow indicators to global warming level. Indicator response is computed as the difference of multi-annual means between end of century (EOC, 2071-2100), middle of century (MOC, 2041-2070), or beginning of century (BOC, 2011-2040) and the reference period (Ref, 1986-2005). Global warming level is computed as the difference in global mean surface air temperature between EOC, MOC or BOC and either the reference period (top axes) or the pre-industrial period (P-I, 1851-1880)(lower axes). Each point corresponds ...... Warming levels of 1.5°C and 2°C compared to pre-industrial are shown with the vertical dashed lines. Regression lines are shown for the response at EOC, MOC, BOC or all three periods (ALL) (except for P). Mean values...."

---

## Author Comment (AC1) · 12 Feb 2018

**Response to Referee 1**

*We thank R1 for this detailed review, which enabled us to significantly improve our article. Enclosed please find a detailed explanation of the revisions we made based on R1's comments. For convenience, comments are in bold and our responses are in italic. Revisions made in the manuscript are presented in italic with grey background.*

**The work by Verfaillie et al. presents a comprehensive analysis of past and future snow conditions at a mid-elevation mountain range in the French Alps. The regional SAFRAN reanalysis and bias-adjusted RCM experiments covering three greenhouse gas emission scenarios are used to drive the Crocus snowpack model, and model simulations are compared against observations at a single measurement site. The snowpack model is employed in a multiphysics ensemble approach which allows for an assessment of the contribution of snowpack modelling uncertainty to the overall projection uncertainty. Results for a range of snow indicators are presented. Concerning the overall future degradation of the snowpack they largely confirm previous works, but also provide a number of new and useful insights that are at least valid for this specific case.**

**Overall, I consider the paper as a relevant and interesting piece of work. The methods and data used are comprehensively described and are well introduced (except for the downscaling and bias-adjustment method ADAMONT, which is however explained in detail in a previous paper). The methodological approach is sound and valid. The introduction and the discussion properly refer to existing works in this field, and the conclusions are well based on the results obtained. There are no language issues, and the topic clearly fits into the journal's scope. As such, I could generally recommend a publication of the work. However, a few minor and one major issues remain, and I'd suggest to ask the authors for a revision of their work in these respects before final publication. Minor issues are listed at the end of this review.**

**The remaining problem with the existing manuscript is its rather technical touch and the wealth of information that is presented in terms of data sets, emission scenarios, scenario periods, methods and especially snow indicators. The comprehensiveness of the work is impressive, but the reader very easily gets lost in this large amount of information that is presented in the text, in the tables and in the figures. These information might be very useful for local stakeholders operating in this very region and being affected by snow conditions, but for a truly useful contribution to the scientific community the results and their presentation need to be much better streamlined in my opinion. The generally most interesting part of the work is probably the entire methodological approach and the possibilities that arise from it. The very detailed results for a representative elevation of 1500 m in the Chartreuse mountain range are more of a case study, and the details of their presentation should receive less emphasis. One option might be to remove parts of the analysis entirely from the manuscript and place it in an additional, accompanying publication (e.g. the multiphysics ensemble analysis which is only briefly described in the results and which has much more potential to be analyzed in more detail). Another option is to move some of the material from the main manuscript to the supplement. This could for instance concern several of the snow indicators (like onset and meltout date), that are in any case only briefly discussed.**

*We thank R1 for the overall positive appreciation of our work and for the suggestions for improvements. Following them, we have decided to move Tables 3 and 5 about STEDs to the supplement as those are only briefly discussed in the text. This will reduce the total number of tables in the main article. We acknowledge the wealth of information provided in this manuscript, and consider worthwile to present both the methods and scientific results from an examp|ary*

*geographical configuration together. We also believe that the geographical location considered (Chartreuse, 1500 m altitude) has broader relevance and that conclusions reached for this case are worth being presented in detail. We agree that several aspects of the present work deserve more in-depth analysis, and this could be addressed by future publications, although some key features can already be analyzed and assessed from the present manuscript*

*While we acknowledge that the content of the manuscript can indeed be considered rather dense, we value this to be a positive quality judgement rather than an issue for a scientific publication, which targets a specialized audience.*

*We also hope that the dense content of our manuscript will be easier to follow after several clarifications in the text. For example, the geographical setting of the case study is now better introduced and separated from the description of the observational datasets (see below). The description of the statistical post-processing was also clarified (see further).*

**In combination with such a streamlining, I'd suggest to put a little more emphasis on the actual processes that are responsible for the identified future changes in the snow indicators. Little is so far said about that. The Crocus model output surely provides an opportunity to do so (e.g., separate analysis of snow accumulation and snow melt amounts).**

*This was already approached in our paper (through the analysis of temperature and precipitation, including the phase of precipitation, as the main drivers of snowpack reduction). While interesting, a further analysis seems out of the scope of our paper, and would make it longer.*

**In this respect, the relation of the snow indicator changes to the GLOBAL temperature change is not very helpful and the authors should think about putting the LOCAL temperature change into focus (though I completely understand the choice of the global scale given political climate targets).**

*Local temperature changes are addressed in detail in the manuscript, in text, tabular and graphical formats. We acknowledge that the reviewer understands our choice to relate local changes in snow indicators to global temperature variations. The reasons and the limitations of this choice are detailed in section 4.4. We understand the reviewer encourages us to relate local changes in snow indicators to local changes in temperature. While doable and certainly leading to significant relationships (at least on 30 years average values) because the physical link is obviously more direct, we preferred not to include this in the revised manuscript in order to not lengthen it and not induce confusions between local temperature and global temperature relationships. This could be addressed in a future study, either by us or other research groups, which may be interested in exploring further results, which could be obtained on the basis of our newly derived dataset..*

**Such a shift of the focus away from details of the case study and towards a more methodological and process-oriented analysis would be very worthwhile in my opinion. Apart from this and as said before, I consider the manuscript as being of high quality and of general relevance for the readership of the journal.**

*We understand the reviewer's point of view, and consider this manuscript to be viewed both as a methodological and application oriented manuscript. We will be pleased to introduce future publications targeting expanded application domains (entire French Alps, Pyrenees, etc.) as well as more in-depth analysis of the drivers for snowpack changes.*

**Minor issues ===== Spatial scale of the Crocus application: What remains somehow unclear is the spatial setup of the Crocus model. I assume the authors use a single-site setup, driven by the outcomes of the SAFRAN reanalysis and of the ADAMONT downscaling method for a representative 1500 m elevation range in the Chartreuse massif. Is that the case? If so, this needs to be clearly said and described in some more detail. It would imply**

**that the results shown are only valid for that specific elevation range in this massif. What about other elevations then? Is it possible to come up with some speculation here as well? Snow projections will surely strongly depend on the elevation considered, and some placement of the results into a broader spatial context would be helpful.**

*Yes, we use a single-site setup (Chartreuse massif at 1500 m as.s.l.). This is now better explained through a new section « 2.1 Geographical setup ». We preferred not speculating on results which could be obtained at other geographical locations and altitudes, because this would unnessarily lengthen an already long manuscript, and will be addressed without speculation in follow-up publications. However, we added the following in the Conclusions : « (…) our results do not directly allow extrapolation of the conclusions in other mountain regions in France or other elevations, although it is expected that the response of neighbouring mountain ranges may be comparable at the same altitude level. ». (p. 30 L. 25-27)*

**Page 1 Line 2: "investigates" instead of "introduces" is probably the better choice.**

*This is now corrected. (p. 1 L. 2)*

**Page 1 Line 9: "reduction in mean interannual snow conditions" is rather unclear.**

*We have rephrased : « reduction in average snow conditions ». (p. 1 L. 9)*

**Page 2 Line 30: "they" instead of "there".**

*This was corrected. (p. 2 L. 29)*

**Page 3 Line 22: The term "currently" is probably wrong. At this point, more GCM-RCM chains are available from EURO-CORDEX. The authors just either specify their date of access of the data base or justify their selection of all available model chains.**

*We used the models available when we last accessed the database to retrieve the data before launching the whole processing chain, and for which the geopotential data for the corresponding CMIP5 GCM (necessary in the ADAMONT method for the calculation of weather regimes) were available.*
*This is now specified : « The 13 GCM/RCM EURO-CORDEX pairs available in April 2017 (and for which the geopotential data for the corresponding CMIP5 GCMs were available) were used. These are expected (...) ». (p. 3 L. 21-22)*
*And also in Section 2.3 : « This study uses the EURO-CORDEX dataset (Jacob et al., 2014; Kotlarski et al., 2014) available in April 2017, consisting of (…). Only the GCM/RCM pairs for which the geopotential data for the CMIP5 GCMs were available were used. » (p. 5 L. 2-5)*

**Page 4 Lines 8-15: Please check: Is SAFRAN really ONLY available over mountain ranges? To my knowledge, entire France is covered.**

*SAFRAN refers here to the original mountain region implementation (Durand et al., 1993). SAFRAN was expanded to wider geographical areas in France (Vidal et al., 2010) and Spain (Quintana-Segui et al., 2017). This is now indicated. (p. 4 L. 21-22)*

**Page 8 Lines 5-21: This method description is rather confusing and very hard to follow. Please streamline.**

*We are sorry that the statistical post-processing appeared confusing, indeed this is a key aspect ouf our work and we attempted to make it clearer in the revised manuscript, which now reads (entire paragraph copied here, p. 7 L. 10-33) :*

*« The entire model chain provides estimates of a series of annual indicators spanning continuously the historical period from 1950 to 2005, typically, to the end of the 21 st century. A total of 13*

*GCM/RCM pairs were considered in the case of RCP4.5 and RCP8.5, out of which 4 are also available for RCP2.6. We generally used a 15-year window to assess the statistical distribution of the indicators considered. For a given GCM/RCM pair and a given RCP, statistics corresponding to a given year can be computed using indicator values for the 15 years surrounding it (7 before, the central year, and 7 after). In what follows, we assume that all GCM/RCM pairs bear equal probability (Knutti et al., 2010). We post-processed the distribution of annual indicator values in two ways.*

*1. Quantiles of annual values: In this case, for a given RCP, all annual values of the indicators spanning the 15 year time window for all the corresponding GCM/RCM pairs were pooled together (195 in the case of RCP4.5 and RCP8.5, 60 in the case of RCP2.6). The quantiles of the distribution of the annual values were determined using a kernel smoothing approach. We computed the 5%, 17%, 50%, 83% and 95% values (Q5, Q17, Q50, Q83, Q95), consistent with IPCC (2013). This approach provides statistical estimates for annual values of the indicator, although it mixes together the effects of interannual variability and inter-model variability.*

*2. Moments of multi-year averages: A running average of annual indicator values was computed using the 15 year sample window, for a given RCP and for each GCM/RCM pair. For a given RCP, mean (μ) and standard deviation (σ) values were computed for the ensemble of multi-annual averages of all GCM/RCM pairs. This approach provides information on the statistical distribution of each indicator for a given RCP on a multi-annual average perspective. In practice, we compute σ' = 0.95 σ, corresponding to the 17% and 83% quantiles in the case of a normal distribution, so that this approach becomes more comparable to the annual quantiles approach described earlier. In the case of the multiphysics Crocus model implementation, we mostly used the multi-year averages approach, and applied it to all Crocus members.*

*The spread of the distributions of these two approaches can be assessed in rather similar ways. In the multi-year average approach, the coefficient of variation CV can be determined as CV= 2 × σ 0 /μ. In the annual quantiles approach, the spread can be assessed by dividing Q83-Q17 by Q50 to form a formal equivalent to the coefficient of variation, defined using quantile values instead of mean and standard deviation (referred to as quantile-based coefficient of variation -QCV-hereafter). ».*

**Figure 1: The STEDx should represent some duration of exceedance and hence need to be represented by some horizontal range in this graph. The representation by single vertical arrows is probably wrong, please check.**

*We agree with this remark and have attempted to improve the graphical representation of this series of indicators.*

**Page 9, line 1: Temperature changes are surely not computed in a relative manner, please check.**

*OK. We changed the sentence to : « Changes were computed (...) ». (p. 9 L. 11)*

**Page 19 Lines 6-10: I assume this is simply an effect of random internal variability at decadal scale, could that be (simulations out-of-phase with reality)? Please clarify.**

*We agree that this is an effect of random internal variability. This is what we wrote in the original manuscript (« low frequency variations at the decadal time scale, superimposing on a long-term trend of general snow reduction »).*

**Page 23 Line 31: Isn't it rather random variations (instead of systematic variations)?**

*We removed the word « systematic ». (p. 25 L. 5)*

**Page 25 Lines 14-16: Is this really the case? Why should a matching of quantile distributions reduce interannual variations? please check and better explain.**

*The method does not reduce interannual variations, but it is clear that it will tend to reduce the spread between different GCM/RCM model results over the calibration time period. We have rephrased to : « which inevitably reduces the spread between different GCM/RCM pairs ». (p. 25 L. 22)*

---

## Author Comment (AC2) · 12 Feb 2018

**Response to Referee 2**

*We thank R2 for this helpful review. Enclosed please find a detailed explanation of the revisions we made based on R2's comments. For convenience, comments are in bold and our responses are in italic. Revisions made in the manuscript are presented in italic with grey background..*

**This paper analyses the mean state and variability in historical and future snow conditions at a mid-elevation location in the French Alps. Analysis is based on output from a physical snowpack model (Crocus) driven by a regional historical reanalysis (SAFRAN) and a suite of historical and future RCM simulations. In situ observations from a nearby location are also used for comparison of the historical conditions.**
**The methodology used in the paper is novel and represents an improvement on previous studies. Results are generally well described compared to existing literature. However I believe there are several changes required and details to clarify before I can recommend final publication.**

*We thank the reviewer for this overall positive appreciation of our work and hope that our revisions and replies will address his/her concerns.*

**Specific Comments:**

**Table 1 obscures the fact that there are only 5 distinct GCMs simulations that sample natural variability. I think this should be pointed out explicitly.**

*We now represent Table 1 as a matrix of the different RCM/GCM combinations.* (p.6)

**Figure 2a-c: In the supplementary material for the RCP8.5 scenario there is a distinct change in the stddev as average snow depth becomes small. It's therefore unfair therefore to suggest that the stddev is stationary based on the RCP4.5 scenario. I think it would be fairer to show the RCP8.5 scenario in the paper and place RCP2.6 and 4.5 in the SI.**

*We agree that RCP 4.5 and RCP 8.5 provide different end-of-century responses of the snowpack for virtually all indicators. It is also true that in the case of RCP 8.5, the snow reduction is sufficiently pronounced that the standard deviation can no longer be considered stationary. However, we believe that it would be unfair to choose RCP 4.5 or RCP 8.5 and focus on either one or the other in the presentation of results and analysis. We intended in the original version of the manuscript to show RCP 4.5 in the main body of the article and provide RCP 8.5 results in the supplement, for the sole sake of saving space. Based on the reviewer comment, we suggest that both RCP 4.5 and RCP 8.5 should be displayed in the main body of the revised article, so that this does not give the reader any impression that we favor one of these two RCPs.*

*The fact that stddev declines through time for RCP 8.5 is now indicated :* « *Figure 2e displays values on the order of 0.08 to 0.11 m with decadal variability but no temporal trend from 1950 to 2100. Figure 3e, on the other hand, shows a decline of standard deviation with time, as SD becomes smaller.* ». *(p. 11 L. 24-26)*

**Figure 2d: The large relative contribution to combined model uncertainty arising from snowpack model multiphysics compared to RCM/GCM inter-model variability seems difficult to reconcile with plots 2a-c (I realize there is interannual variability in Figs 2a-c, while much of this signal is removed in the multi-annual average). Is it possible to present**

**15 year running means for the 13 RCM/GCM tracks shown in Fig 2c or at least for a representative subset of the 13 pairs along with Fig d (either in this same figure, or in a separate figure similar to how you subsequently separate quantile plots based on annual frequency data and multi-annually averaged data)? This may allow the reader to get a better sense of the relative spread in the ESCROC ensembles compared to the RCM/GCM inter-model variability.**

*We agree that the relative contribution of uncertainty displayed by Fig. 2d (2e in the revised manuscript) could not be directly associated with the spread of Fig. 2c because the 15-year running mean removes the high interannual variability. Therefore, we added an intermediate panel (Fig. 2d) which is simply the 15-year running mean of Fig 2c. We think that this helps the reader understand all the post-processing and the links between the different subplots.*

*The description of this Figure was modified accordingly in section 3.1 :*
*« Figures 2d and 3d present the 13×35 15-year running average values spanning all simulation members of Figures 2c and 3c respectively. This corresponds to the second statistical post-processing described in Section 2.5.2 which removes the interannual variability and allows an easier quantification of each source of uncertainty.*
*Figures 2e and 3e aim at apportioning the uncertainty in the time series of Figures 2d and 3d respectively, between the uncertainty arising from GCM/RCM inter-model variability (including model uncertainty and internal variability of climate at different time scales) and the uncertainty arising from the multiphysics snowpack model. For that purpose, the standard deviations of the 455 values of Figures 2d and 3d were computed for each 15-year window, and correspond to the total standard deviations of the SD. This is shown in black solid line in Figures 2e and 3e. (...) ».*
*(p. 11 L. 17-24)*

**Further to this point — in the text there are several times that you refer to the relative fraction of uncertainty contributed by snowpack model multiphysics as ~20%, yet this graph shows that is can be as high as 80%. Please justify the use of 20%. Is there a rationale for why the uncertainty due to snowpack model multiphysics is higher in the historical period, even though the combined model uncertainty remains fairly constant?**

*R2 is right that the value of 20 % uncertainty is only valid for future time periods. The answer concerning the fact that uncertainty due to snowpack model multiphysics is higher in the historical period is partly given p. 12 L. 1-3 : « (...) the historical period is affected by the varying number of available GCM/RCM before 1980 and by a potentially artificial reduction of spread over the 1980-2011 calibration period of the ADAMONT statistical adjustment method ».*

*Just after this sentence, we added this implication in the manuscript:*
*« This could partly explain why the uncertainty of GCM/RCM appears lower than the multiphysics uncertainty on the historical period, in combination with the deeper snowpacks in the historical period. ». (p. 12 L. 3-4)*

*Indeed the uncertainty also declines with time linked to increased snow scarcity, as already indicated (p. 11 L. 32-35): « The ESCROC component shows values ranging from 0.02 m to 0.07 m, exhibiting rather smooth fluctuations from 1950 to 2100 and a general decreasing trend, along with the general decreasing trend of SD over the considered time period (see below) ».*

**Figure 3 and 4: The quantiles from different RCPs overlap too much in these figures to discern one set of shading from the others. I suggest showing RCP8.5 only in these figures. The results from the other RCP scenarios are provided in tabular form in the main document which I think is sufficient (along with the plots of other RCPs in the SI).**

*We do not agree to show only one RCP in the main text, and prefer to show all of them together, with individual RCP plots in the supplement. There is no rationale for choosing either one the existing RCPs.*

**Figure 3: Why do the SOD and SMOD for RCP8.5 begin to encompass summer months? Is this an error with the calculation?**

*We thank R2 for this remark. Indeed, there was an error in the calculations of SOD and SMOD for years in which SD was never greater than 5 cm. The algorithm was corrected with implications for Figs. 4-5 and S3 & S6 , as well Tables 2 and 3.*

**P8.L15-21: Please clarify why you use a distance of +/- 1.37\*sigma from the mean for the 17th and 83rd percentiles? Shouldn't the 17th and 83rd percentiles be 0.95 sigma away from the mean such that CV=1.9\*sigma/mu?**

*We thank the reviewer for spotting this error. Indeed, the Q17 and Q83 percentiles correspond to ± 0.954 sigma distance to the median, we used an erroneous value in the original submission. All tables, graphics and text using Q17 and Q83 percentiles values from multi-annual average values have been updated accordingly. We are particularly grateful to the reviewer to have identified this error, which may have caused propagation of erroneous indications of variability and spread from our projections had it not been corrected during the review process.*

**P25.L28-29: Please rephrase in order to account for the relative component attributed to snowpack modeling errors in both future and historical periods.**

*This was rephrased : « (…), under the conditions of the Northern French Alps and after the middle of the 21st century, the uncertainty component attributed to the snowpack modeling errors alone is on the order of 20%, (...) ». (p. 25 L. 33 to p. 26 L. 1)*

**P27.L10: This statement depends on the RCP scenario. It is not the case for RCP8.5.**

*This is now clarified: «(…) and even increases in relative terms (until the middle of the century for all RCPs, and towards the end of the century for all RCPs except RCP8.5), (...) ». (p. 27 L. 13-14)*

**P28.L15-17: Could there also be a change in the mean density of snowfall occurring at the location?**

*The density of snowfall depends on temperature and wind speed during the snowfall. While this is an interesting hypothesis to test, we consider this to be beyond the scope of this article and to be addressed in a future study.*

**P29.L8-10: Please reword or justify the claim that snowpack modeling uncertainty is typically 20% when Figure 2d shows it can be up to 80%. I agree that it may have a smaller impact on trends.**

*The value of 20 % is valid only in the future. This sentence was rephrased : « Uncertainty arising from physical modeling of snow after the middle of the century can account to 20% typically of the simulation results ». (p. 29 L. 25-26)*

**P29.L11-17: The ADAMONT method was evaluated in a previous paper. Please be clear as to which aspects of these conclusions were accomplished in this paper.**

*The ADAMONT method was described and evaluated only on one RCM driven by a reanalysis in Verfaillie et al., 2017. Here we apply it to the EURO-CORDEX RCM/GCM ensemble spanning the historical period and future projections for the 21st century. This is the first published use of the method, so that it is the first evidence of the capability of the method to be used to adjust a large number of regional climate model results and provide consistent meteorological forcing data for the land surface model Crocus.*

**Further to this point, while you argue that your methodology is an improvement to previous studies that use delta change methods, your assessment of the results says that they, in**

**fact, agree with these previous studies. Under what circumstances might you expect to see differences? Is it possible to provide a direct comparison between your methodology and a delta change method for this location or to highlight statistics that would differ between the two methods?**

*We agree with the reviewer that our results are consistent with results obtained using delta-change methods, in French mountain regions as well as in Switzerland, as quoted in the manuscript (e.g. Castebrunet et al., Schmucki et al.). However, this consistency is only demonstrated for multi-annual multi-model trends on snow depth or snow water equivalent mean values. We strongly believe that our model chain, using the RCM chronology, will capture more appropriately potential changes in timing of precipitation, and as such could only be compared to raw output of RCM (e.g. Steger et al.). Differences would be expected under a situation where the chronology of precipitation would differ significantly in the future, because the delta-change approach would only modify the air temperature and rain/snow partitioning, but not the timing of the events. These changes in the multivariate chronology of meteorological events in the Alpine region have not been investigated in details until now to the best of our knowledge although their stationnarity is a requirement for the validity of the delta-change method. Furthermore, although our results do not exhibit significant changes in the interannual variability of the snow indicators, this is a result of our projections whereas it is only an assumption in the delta-change method.*
*While this could be interesting to demonstrate whether results obtained using delta-change approaches could still be employed for impact studies, we do not feel the need to perform such comparisons ourselves given that there are no arguments supporting that our approach could provide less appropriate results than a delta-change approach. We have no ressource to implement a delta-change method for the purpose of such a comparison, now that we have developed and implemented the full model chain described in this manuscript. We are, however, fully eager to communicate our data to other research groups interested in performing such a comparison in the future.*

*We added a paragraph at the end of Section 4.3 to explain this in more details :*
*« Many of the results discussed above indicate a strong consistency between our results and results obtained using delta-change methods, in French mountain regions as well as in Switzerland (e.g., Castebrunet et al., 2014; Schmucki et al., 2014). This consistency is shown for multi-annual multi-model trends on snow depth or snow water equivalent mean values, but cannot be assessed regarding the interannual variability because this is generally not addressed in these studies. The model chain implemented here, explicitly making use of the intra-seasonal and inter-seasonal RCM chronology, inherently captures more appropriately potential changes in timing of precipitation. Differences between the current study and studies based on delta-change approaches would be expected under a situation where the chronology of precipitation would differ significantly in the future, because the delta-change approach would only modify the air temperature and rain/snow partitioning, but not the timing of the events. These changes in the multivariate chronology of meteorological events in the Alpine region have not been investigated in details until now to the best of our knowledge, although their stationarity is a requirement for the validity of the delta-change method. Furthermore, although our results do not exhibit significant changes in the interannual variability of the snow indicators, this is a result of our projections whereas it is only an assumption when applying a delta-change method. More in-depth comparisons between outputs of delta-change approaches and direct adjustments to RCM output could be carried out in the future, but are beyond the scope of this article. ». (p. 27 L. 19-32)*

**Technical corrections/Suggestions:**

**I find the use of the phrase "annual-scale" a bit unnatural. I suggest using "annual indicators" as you occasionally do (P8.L15) throughout the paper.**

*This was corrected. (p. 1 L. 2)*

**Similarly, there are places in the paper where you might consider replacing the word "variation" with "change", "response", "difference" or "variability", but I'm having trouble articulating a clear rule to follow in this. Variation is frequently reserved to specify a very small adjustment.**

*OK. We corrected this throughout the manuscript.*

**Title: "Multi-component ensembles. . .."**

*The title was changed accordingly.*

**P1.L2: "This article investigates the climatic response of a series of indicators for characterizing annual snow conditions and corresponding meteorological drivers at 1500 m altitude in the Chartreuse mountain range in the Northern French Alps. "**

*OK. This sentence was modified as suggested by R2. (p. 1 L. 2-3)*

**P2.l30: "because they are newer. . .."**

*Done. (p. 2 L. 29)*

**P3.L2-4. Please rephrase these two sentences to make the typical delta-change approach clearer.**

*We rephrased : « (…) a pre-determined difference (delta) of temperature and/or precipitation values to an observation record, based on changes computed using climate models (either global or regional). This cannot capture combined changes in temperature, precipitation and other meteorological factors, in terms of magnitude of the fluctuations and their seasonal-scale and interannual variability. ». (p. 3 L. 1-3).*

**P8.L15: "Moments of multi-year averages: A running average of annual indicator values is computed (typically with a 15 year sample window), for a given RCP and for each GCM/RCM pair."**

*We changed the sentence to: « Moments of multi-year averages: A running average of annual indicator values was computed using the 15 year sample window, for a given RCP and for each GCM/RCM pair. ». (p. 7 L. 23-24)*

**P9.L1 "for 15-year windows around each future time period t and each RCP r"**

*We changed this sentence to : « for 15-year windows around each future time period t for the RCP r ». (p. 9 L. 12-13)*

**P9.L3: "i.e." in place of "e.g."**

*Done. (p. 9 L. 15)*

**P10.L19: "It highlights the significant interannual variability in observed, reanalyzed and climate model datasets."**

*Done. (p. 10 L. 27)*

**P11.L37: "which highlights the need for appropriate data synthesis methods". Please elaborate.**

*We added the following sentence : « Indeed, it is not possible to draw conclusions or make decisions on the sole basis of such a raw ensemble of individual scenarios. ». (p. 11 L. 15-16)*

**P21.L10: to widen**

*This was corrected.* (p. 21 L. 4)

**P23.L27: "By definition no performance metrics pertaining to annual variations can be computed between the adjusted climate output and either observations or reanalysis data, because the two are not designed to exhibit synchronous variations."**

*The sentence was corrected : « By definition no performance metrics pertaining to annual fluctuations can be computed between the adjusted climate output and either observations or reanalysis data, because the two are not designed to exhibit synchronous fluctuations. » (p. 25 L. 1-2)*

**P25.L4: "independent from the time period for calibration of the ADAMONT adjustment method (1980-2011). . .."**

*OK, this was corrected. (p. 25 L. 10-11)*

**P25.L9-10: "or applying the final quantile mapping separately to rain and snow precipitation in order to mitigate detrimental interactions between temperature and precipitation (Verfaillie et al., 2017). . .."**

*This was corrected. (p. 25 L. 16-17)*

**P25.L30: "Because the number of GCM/RCM model pairs was different for RCP2.6 (4) and RCP4.5 and RCP8.5 (13), we compared the statistics for indicators during the historical period based on the 4 RCP2.6 pairs alone, as well as the full ensemble of 13 GCM/RCM pairs."**

*Done. (p. 26 L. 3-5)*

**P26.L11: "similar statistics are found for these 4 model pairs as for the full ensemble of thirteen."**

*Done. (p. 26 L. 15-16)*

**P26.L24: I'm not sure what you mean by "snow-dry" seasons. Seasons without snow on ground or without snowfall occurring at this location at all?**

*We meant seasons without snow on the ground. This is now better explained: « (…) and more frequent seasons with barely any snow on the ground ». (p. 26 L. 28)*

**P26:L26-28: "The decreasing SD trend is also combined with a decreasing SWE trend (~ -6 kg m−2 per decade for RCP2.6, -18 kg m−2 per decade for RCP4.5 and -35 kg m−2 per decade for RCP8.5 over the period 2030-2090, Table 4) and a decreasing trend in duration of STED5 (as in Marty et al. (2017a)), STED50 and STED100 (Table5)."**

*We have replaced the sentence by :*
*« The decreasing SD trend is also combined with a decreasing SWE trend (~ -6 kg m−2 per decade for RCP2.6, -18 kg m−2 per decade for RCP4.5 and -35 kg m−2 per decade for RCP8.5 over the period 2030-2090, Table 3) and a decreasing trend of STED5 (as in Marty et al. (2017a)), STED50 and STED100 (Table S2). ». (p. 26 L. 30-33)*

**P27.L33: "This is all the more relevant in that none of the GCMs used for this study. . .."**

*Done. (p. 28 L. 18-19)*

**P28.L6-8: ". . .in contrast to previous studies (Durand et al., 2009a; Pepin et al., 2015). This result may stem in part from the fact that although elevation dependent warming is generally maximal in the fall and springtime, our target period covers mostly wintertime. Alternatively, this low enhancement factor could be due. . ..”**

*This was corrected (p. 28 L. 26-28).*

**P28.L26-28: "The multi-component ensemble framework makes it possible to account for the various sources of uncertainty and variability that affect future climate projections, some of which are neglected in both previous and ongoing climate change impact studies.”**

*Done.(p. 29 L. 12-14)*

**P28.L28-32: Split into more than 1 sentence.**

*The sentence was split in two: « The multi-ensemble framework developed here draws on several RCPs (RCP 2.6, RCP 4.5 and RCP8.5), feeding several GCM model runs from the CMIP5 intercomparison exercise, which themselves feed various RCP model runs from the EURO-CORDEX downscaling exercise. Those are adjusted using the refined quantile mapping method ADAMONT against the meteorological reanalysis SAFRAN, making it possible to drive a multi-physical version of the energy balance multi-layer snowpack model Crocus. » (p. 29 L. 14-18)*

**P28.L32-36: "The method defines a series of annual snow and meteorological indicators that represent various aspects of winter snow conditions. . ..”**

*We changed the sentence to :*
*« The method defines a series of annual snow and meteorological indicators that represent various aspects of the winter season (...) ». (p. 29 L. 18-19)*

**P29.L21: "exhibit similar statistics at the interannual and multi-annual scale as the full 13-member ensemble, . . ..”**

*OK. (p. 30 L. 5-6)*

**P29.L26: "but maintained interannual variability. . .. . . “ Rephrase.**

*We changed the word « maintained » to « sustained ». (p. 30 L. 11)*

**P29.L28-29: "As assessed in this study, for this location, interannual variability is larger than inter-model spread for a given RCP scenario.”**

*This was corrected. (p. 30 L. 13-14)*

**P29.L32-33: "the latter leading to frequent occurrence of ephemeral or nearly snow-free conditions at the end of the century.”**

*OK. (p. 30 L. 17-18)*

**P29.L35: "For example, the change in mean snow depth”**

*OK. (p. 30 L. 20)*

**P30.L4: "this value changes very rapidly”: I dislike the wording that it changes "rapidly" since the changes on Figure 5 are quite linear (except for STED100). Please rephrase. I suggest something along the lines of "the magnitude consistently increases along with global mean temperature reaching reductions of 80% beyond 4 °C of global warming.**

*OK. We have chosen the formulation suggested by R2. (p. 30 L. 22-23)*

**P30.L8-10: "These locations may be investigated in the future, based on the methodological framework introduced here and the data available in the SAFRAN reanalysis for the French Alps and Pyrenees (Durand et al., 2009b, a; Maris et al., 2009)."**

*OK. (p. 30 L. 27-29)*

**Figure 2 Caption: "c) Ensemble of Crocus model configurations driven by the 13 RCP4.5 GCM/RCM pairs; each GCM/RCM pair is displayed with a different color."**

*OK. The Figure caption was changed accordingly. (p.13)*

**Figure 4 Caption: "Ensemble spread in 15-year running mean ( µ ± σ' ) of all GCM/RCM pairs for each scenario (HIST, RCP2.6, RCP4.5 and RCP8.5), along with 15-year running means of observations (1960-2016) and SAFRAN-Crocus runs (1958-2016) at CDP, for: . . .."**

*OK. This is now Figure 5. The Figure caption was changed accordingly. (p. 18)*

**Figure 5 Caption: "Response of local meteorological and snow indicators to global warming level. Indicator response is computed as the difference of multi-annual means between end of century (EOC, 2071-2100), middle of century (MOC, 2041-2070), or beginning of century (BOC, 2011-2040) and the reference period (Ref, 1986-2005). Global warming level is computed as the difference in global mean surface air temperature between EOC, MOC or BOC and either the reference period (top axes) or the pre-industrial period (P-I, 1851-1880) (lower axes). Each point corresponds . . .. . .**
**Warming levels of 1.5 °C and 2 °C compared to pre-industrial are shown with the vertical dashed lines. Regression lines are shown for the response at EOC, MOC, BOC or all three periods (ALL) (except for P). Mean values. . .."**

*OK. This is now Figure 6. The Figure caption was changed accordingly. (p. 23)*

---

## Author Response (AR2)

Dear Editor,

We thank you and the two reviewers for additional remarks and suggestions on our manuscript. We have analyzed them and provide below point-by-point replies. We also provide a revised manuscript incorporating our proposed changes to the manuscript. We hope that you will find this version suitable for acceptance and publication.

Yours sincerely,

Samuel Morin, on behalf of the author team.

Reply to editorial comments:

Note that comments from the Editor Ross Brown and reviewers are in plain text, our replies are italicized.

Comments to the Author:

Dear Authors - I have received the referee reports of your revised manuscript and am pleased to accept your paper for publication subject to your response to the corrections and comments noted by the two reviewers. Both reviewers noted major improvements in the m/s but Reviewer 2 has some remaining concerns that I would like you to respond to (for Editor review). I look forward to your response.

We thank you and the reviewers for the positive appreciation of our work, please find below specific replies to the comments and suggestions.

In reading your Conclusions I noticed some minor language issues that I've itemized below. Reviewer 1 also mentioned finding the writing style difficult to follow in places so it would be good if you can get a technical editor to go through the paper before submitting the final version.

We have checked once again the content of the manuscript and have fixed the remaining grammar and style issues, to the best possible extent. We also took into account the fact, that accepted paper currently undergo a comprehensive copy-editing step before final publication, and our article will certainly benefit from it as part of the full publishing process in The Cryosphere.

1. Note missing "it" on p. 30 line 8 "... we consider [it] preferable to ...'

Done.

2. p. 30 lines 10-13: This sentence is incomprehensible as written. I suggest you split this into two bullets addressing (1) the trajectory of changes with RCPs, and (2) interannual variability. The next bullet starting in line 15 relates to (1) so you can consider consolidate the RCP-related conclusions in one bullet.

We have simplified and rewritten the following sentence :

"Ensembles of climate projections generated under RCP2.6, RCP4.5 and RCP8.5 are rather similar until the middle of the 21st century, with the continuation of the ongoing reduction in mean interannual snow conditions, but sustained interannual variability of snow conditions, playing even an increasing relative role along with the decrease of mean snow conditions."

to

"Projections of meteorological and snow conditions corresponding to RCP2.6, RCP4.5 and RCP8.5 show similar behaviour until the middle of the 21st century. They all exhibit significant interannual variability, and a long term trend of increasing snow scarcity".

We have further modified the following sentence :

"As assessed in this study, for this location, interannual variability is larger than inter-model spread for a given RCP scenario."

to

"Our study shows that, for this location, the interannual variability is larger than inter-model spread for a given RCP."

We kept the organization of the bullets similar, the first one relating to the first half of the 21st century, the second one to the second half.

3. p. 30 line 19: "...show significant correlation[s] with ..."

Done.

4. p. 30 line 21: "... for [a] 1.5C global temperature..."

**Done.**

5. p 30 line 33: consider replacing "...making it possible to compute Crocus snowpack model runs able to tackle avalanche hazard evolution,..." with " making it possible to simulate the evolution of avalanche hazard with Crocus, ..."

We have replaced "making it possible to compute Crocus snowpack model runs able to tackle avalanche hazard evolution" by " making it possible to address the impact of climate change on avalanche hazard using Crocus model runs".

6. p 31 lines 8-10: consider simplifying the phrase "Conversely, it is re-emphasized here that, while changes in natural snow conditions as projected in this work are likely to affect operating conditions of ski resorts, no quantitative conclusions can be made, given that snow management practices induce significant changes in operating conditions, which depend on intricate[d] factors related to temperature and precipitation..."

Suggestion ...

"It must be emphasized that while the projected changes in natural snow conditions shown in this work are likely to affect operating conditions of ski resorts, no quantitative conclusions can be made as snow management practices, especially the production of artificial snow, depend on an intricate set of temperature and precipitation related sensitivities..."

We have replaced "Conversely, it is re-emphasized here that, while changes in natural snow conditions as projected in this work are likely to affect operating conditions of ski resorts, no quantitative conclusions can be made, given that snow management practices induce significant changes in operating conditions, which depend on intricate[d] factors related to temperature and precipitation.."

**by**

"It must be emphasized that while the projected changes in meteorological and natural snow conditions shown in this work are likely to affect operating conditions of ski resorts, no quantitative conclusions can be drawn on this topic. Indeed, snow management practices, especially snowmaking, play an essential role in their operations, and they should be accounted for in studies specifically addressing the impact of climate change on this socio-economic sector (Hanzer et al., 2014, Spandre et al., 2016, Steiger et al., 2017)."

7. p 31 lines 12-13: The follow phrase is a rather bizarre and unnecessary "... mechanistic derivations of the impact of meteorological conditions on the socio-ecosystemic compartment under investigation". I suggest you remove it.

We have replaced "Beyond these applications to seasonal snow, the method is ready to use for

hydropower potential, water resources assessments, glacier mass balance studies, ecology, natural hazards related to meteorological conditions and more generally environmental impact studies which can be based on mechanistic derivations of the impact of meteorological conditions on the socio-ecosystemic compartment under investigation."

by

"Beyond these applications to seasonal snow, the method is ready to use for a wide range of environmental impact studies addressing various mountain features potentially affected by climate change, such as natural hazards, cryospheric components (glaciers and permafrost), water resources including hydropower, ecosystems functioning and the impact of their changes on human societies."

**Reviewer #1**

I appreciate most of the changes introduced by the authors into the revised version of their manuscript. Moving Tables 3 and 5 to the supplement certainly streamlines the presentation.

We thank Reviewer #1 for his/her positive appreciation of our revised manuscript, please find below a point-by-point reply to his/her comments and suggestions.

However, I'm sorry to say that the replies to my few major comments and the respective revisions are not satisfying in my opinion. As said before, the paper is a very valuable piece of work. But the amount of information presented is (too?) large, and physical interpretations of the results are largely missing. Also the representativeness of the work and its transferability, as suggested by the title of the manuscript that takes a very broad perspective, is not yet addressed convincingly. Yes, the paper has a length issue, but rebutting required improvements with the only argument of length constraints, although several suggestions were provided to cut down the length, is not convincing to me. Please find a summary of these issues below. I believe incorporating at least some of the suggested changes would strongly enhance the quality and the significance of the work. As the paper itself is a valid contribution, I'd leave it up to the editor to decide if further revisions are necessary or not.

With kind regards.

We thank Reviewer #1 for his/her suggestions We have performed further amendments to our manuscript, and we hope that they alleviate the concerns raised above regarding the two main points described below.

Representativeness: As outlined in my first review, a broader geographical validity of the results is not given per se. This is especially true for elevation-dependent processes. The point-scale application for a specific elevation in a specific massif is valuable, but it is a case study as long as no arguments for a broader significance are provided. Neither in their replies nor in the paper itself the authors provide such arguments. They claim a broader applicability of the results, at least for similar altitudinal levels, but without providing their lines of thought. This would be fine as long as this restriction would be properly reflected by the paper's title. However, the current title implies a much broader significance ("in the Northern French Alps") than is currently provided.

We understand this concern, and we have addressed it in two ways. We suggest to add "(Chartreuse, 1500 m altitude)" to the title, thereby highlighting that while the geographical region addressed in this publication is indeed located in the Northern French Alps, our results specifically apply to a given altitude and geographic location. We believe that the abstract makes it very clear regarding the geographical set-up of this study.

We have provided more evidence to justify that the local impacts of climate change in neighboring massifs at the same elevation range should be similar, and have modified and introduced material in the following part of the conclusions section :

"While this work provides scientific results directly exploitable for snow and meteorological conditions at 1500 m altitude in the Chartreuse mountain range, our results do not directly allow extrapolation of the conclusions in other mountain regions in France or other elevations. It is, however, expected that the response of neighbouring mountain ranges may be comparable at the same altitude level, because their behaviour in the past (Durand et al., 2009a, 2009b) and in previous studies addressing future changes (Rousselot et al., 2012, Castebrunet et al., 2014) was generally rather similar. This remains to be explored more quantitatively and will be the topic of upcoming studies, based on the methodological framework introduced here and the data available

in the SAFRAN reanalysis for the French Alps and Pyrenees (Durand et al., 2009a, 2009b, Maris et al., 2009)."

Physical processes: Also in the revised version, the processes that are actually leading to the projected snow cover changes and that are associated with different kinds of uncertainties are basically not discussed. The mentioned analysis of temperature and precipitation phases is reflected by as few as 3 to 4 sentences in Chapter 3.3.

We take note of the reviewer suggestion, and argue that it is not the purpose of the current manuscript to discuss in detail the physical processes responsible for the change in mountain snow conditions under climate change. We have, however, added a paragraph in section 4.3, which reads:

"The comparison of trends of meteorological indicators (temperature, total precipitation and ratio of snow to total precipitation) and indicators characterizing the state of snow on the ground provides insights into the physical mechanisms responsible for changes in snow conditions. The snowpack is progressively initiated and complemented by precipitation events during the wintertime, and it is thus unsurprising and consistent with previous evidence that the decline in snow precipitation is one of the main responsible for the decline in snow conditions, even if total precipitation does not exhibit any significant trend (Steger et al., 2013, Gobiet et al., 2014, Castebrunet et al., 2014, Lafaysse et al., 2014, Schmucki et al., 2017a, Beniston et al., 2018). That the reduction of the snow season is asymmetrical with a stronger reduction in the spring than in autumn is consistent with the fact that not only snow precipitation amounts drive the response of the snowpack to climate change, but also the intensity of the melt rate, which also depends on atmospheric conditions and is enhanced under warmer conditions (e.g., Steger et al., 2013, Pierce and Cayan, 2013). The data sets underpinning the present study could be used to address in a more quantitative manner the physical processes responsible for the results of the simulations, however this falls beyond the scope of this study (Pierce and Cayan, 2013)."

The arguments of the authors to focus on the relation to GLOBAL temperature changes are not convincing in my opinion. First of all, the underlying processes should be identified. A relation to global warming targets could be a follow up, but not the other way round.

*We have modified the introductory paragraph to the discussion on this point (section 4.4), which now reads:*

"The international framework for climate negotiations, culminating at the yearly Conferences Of Parties (COP), and basing the technical part of its decision process on IPCC assessments, shows a strong tendency to focus on global temperature changes. In recent years, there has been increasing societal demand for quantifying the local impacts of global warming levels since the pre-industrial time period of 1.5°C, 2°C and beyond."

Indeed, the scientific community is encouraged to produce the evidence, if it exists, on the links between global warming levels and local impacts. Other similar types of studies have emerged in the literature recently (e.g., Kraaijenbrink P.D. A., et al., Impact of a global temperature rise of 1.5 degrees Celsius on Asia's glaciers, Nature, 549, 257–260, doi:10.1038/nature23878, 2017) for Himalayan glaciers, and it is useful that similar types of assessments are performed for the mountain seasonal snowpack. In addition, performing such experiments, as outlined by literature quoted in our manuscript (e.g. James et al., 2017), does not necessarily require to identify physical linkages between global temperature levels and the response predicted by the cascade of models used for downscaling and impact assessments. We believe that our findings of a significant correlation between all local indicators of snow conditions (averaged over 30 years), and the corresponding global warming level, is a novel and significant result which deserves to be brough to the attention of the scientific community. We have added an additional discussion sentence at the end of section 4.4, regarding the likely physical reason behind this behaviour, although we feel that the result could stand alone even without this interpretive statement.

"Nevertheless, the significant correlation between 30-years average global temperature difference to pre-industrial levels of the GCMs, and the local effects on air temperature and snow conditions simulated using the same driving GCMs processed by means of a cascade of physically-based (RCM) and statistical (ADAMONT) downscaling and adjustment methods, followed by the use a multi-layer energy and mass balance snowpack model (Crocus), is consistent with the fact that (i) 30-years average regional and local temperature in the European Alps are strongly and directly influenced by the global climate and (ii) the multi-annual mean response of the snowpack at 1500 m altitude is substantially governed by and responds to multi-annual mean local air temperature."

**Reviewer #2**

We thank Reviewer #2 for his/her addition review of our manuscript. Please find below a point-bypoint reply to his/her suggestion.

A few additional technical corrections I noticed: pg. 3, L11: "carefully handling"

Done.

pg. 11, L10: scenarios

Done.

pg. 11, L20: "Figures 2e and 3e aim to apportion ... "

Done.

pg. 12, L4: "during the historical period, in combination..."

Done.

pg. 26, L32: "and decreasing trends of STED5...."

Done.

[revised manuscript text omitted]